# Karst Ecosystems of Middle Timan, Russia: Soils, Plant Communities, and Soil Oribatid Mites

Elena N. Melekhina * , Vladimir A. Kanev and Svetlana V. Deneva

Institute of Biology, Komi Science Center, Ural Branch of Russian Academy of Sciences (IB FRC Komi SC UB RAS), Syktyvkar 167982, Russia
* Correspondence: melekhina@ib.komisc.ru

**Abstract:** Oribatid mites are distinguished by high taxonomic diversity and abundance in almost all natural zones; they are used as an indicator group of microarthropods in the study of natural landscapes and anthropogenically disturbed ecosystems. In the karst landscapes of the North, the oribatid mites are very poorly studied. The aim of this study was to investigate the diversity of soil types, plant communities, and oribatid mites in karst relief forms in the conditions of the northern taiga forests. The material was collected in July 2020 in the karst landscapes of Timan Ridge, found in the European northeast of Russia. The research sites were located in the profile of the karst crater and in the profile of the slope in the Ukhta River Valley. A sedge wetland community, and pine–bilberry–green-moss forest, located in the depression between glacial hills and uplands were also examined. A total of seven sites were examined. Geobotanical descriptions were made by standard methods, descriptions of the soil profile, and samples of the soil microfauna; 12 in each site were collected. A total of 51 oribatid mite species from 39 genera and 31 families were found. The highest taxonomic diversity of oribatids was noted in forest phytocenoses located in the upper part of the karst crater slope, the lower part of the karst crater slope, and on rock outcrops in the lower part of the slope in Ukhta river valley. Ordination of the oribatid mite community by NMDS method showed the association of sites S3, S4, and S5 located on the slope of the karst crater in one group, and sites S6 and S7 located on a slope in the Ukhta River Valley, as well as S2 (pine–bilberry–green-moss forest located in the depression between glacial hills and uplands), in another group. The swamp community was located separately from other communities. Species of oribatid mites, which created the specifics of each community, were noted. The specifics of the population of oribatid mites of karst landscapes were that along with the features of fauna, characteristic for zonal north-taiga forests (the predominance of polyzonal widespread species) were found the "conditionally southern" species, the main area of distribution of which is located in lower latitudes. The study provides the basis for future studies of poorly known oribatid mites of karst landscapes of Northern Europe.

**Keywords:** karst craters; regosols; leptosols; cambisols; phytocenosis; Oribatida; european northeast

## 1. Introduction

Karst landscapes within geographical zones are often places where intrazonal types of soils and vegetation, as well as areas of specific biota are located [1]. The key features of modern karst landscapes were established in the postglacial period [2]. Karst reliefs are formed by groundwater activity on land areas, the surface of which comprises soluble rocks, characterized by a variety of surface and underground objects [3–6]. Among the numerous forms of surface karst, deep, narrow, cone-shaped sinkholes are notable.

Under conditions of a cold humid climate, on the parent rocks of glacial genesis prevailing in the taiga of the European part of Russia, zonal soils are formed in autonomous positions—podzols, podzolic, etc. [7,8]. The series of autonomous mesomorphic soils significantly expands in karst areas, where older—in particular, Permian—carbonate and sulfate rocks come to the surface from under the cover of quaternary deposits [9,10].

Researchers of soils in karst regions agree that the soils and soil cover of these territories have a low potential for restoration, and the use of soils for economic purposes can lead to catastrophic and, in some cases, irreversible consequences [11–14]. The most valuable areas for the conservation of biological diversity in the European northeast are included in the system of specially protected natural territories of the Komi Republic [15–17].

Soils formed on the products of weathering of dense carbonate rocks are among the least studied in the northeast of the European part of Russia. Comparative analysis of literature data covering various natural zones reflects significant differences of the taiga soils of continental regions on carbonate rocks and their analogues in the humid (subgumidic) areas of moderate continental climate [18–24].

Floristic studies were carried out mainly in areas where there are outcrops of bedrock carbonate rocks, such as limestones and marls [25]. Studies were carried out in complex reserves, "Belaya Kedva" and "Pizhemsky", on the Middle Timan, where karst landscape forms are widely represented [26–28]. Here, the state of coenopopulations of rare, protected plants in karst landscapes was studied. In the adjacent territories, in the Arkhangelsk and Vologda regions, these works were carried out on karst landforms, in particular on the White Sea–Kuloi plateau, where the flora of such areas and their roles in the preservation of rare relic floristic complexes were revealed [2,29].

According to modern classification, the area of the Tobys and Ukhta river basins belongs to the Kanino–Timan karst province of the East Timan karst area of the Russian Plain [30]. The most intensive karst processes on Timan are observed within Middle Timan in the areas of shallow carbonate deposits of the Lower Permian and Carboniferous periods. The age of this karst is considered by most researchers to be modern. The degree of karstification of the area is weak or significant, whereas it is high and extreme in some sections [31].

The Tobys river basin contains carbonate and sulfate karsts. The carbonate karst is developed from limestone and dolomites. In areas with karst development, karst craters are widespread, with their diameter and depth sometimes reaching 20–30 m and 10–15 m, respectively. Formation of karst craters and hollows at the confluence of craters is not uncommon. Some karst craters are filled with water, forming karst lakes. Some lakes are formed in the place of "merged" karst craters. The density of the karst forms typically does not exceed 10–20 per km$^2$, but in the basins of the rivers Sedju and Tobys and upstream of the Ukhta River, the density can reach 60–100 per km$^2$ [30]. Sulfate karsts in the Tobys River basin are represented by the Kungurian Stage.

The main karst forms in the areas are corrosion–suffusion craters of different sizes and groups. On the left bank of the Tobys River, which is strongly affected by karst, in addition to the usual craters, there are 20–30 m-deep gaps with steep walls, in which gypsum and anhydrite are exposed [30]. Glaciation was of great importance in the development of the Timan relief. The river valleys comprising sediments of glacial origin and with relatively easily eroded bedrock are not wide and have well-developed accumulative, less frequently incised terraces. The greatest karsting is observed in watersheds and the second floodplain terrace, where the thickness of quaternary sediments does not exceed 10–15 m [30].

We suggested that on karst forms of relief, specific soils are formed with a special structure of the soil profile and a specific chemical composition. Accordingly, one could expect a specific composition of plants and soil invertebrates. As a representative group of soil invertebrates, oribatid mites (Oribatida) were selected. Oribatid mites are distinguished by high taxonomic diversity and abundance in almost all natural zones and are present in all soil types. Oribatids respond to changing habitat conditions by changing the abundance, diversity, and structure of groups. Oribatids are used as an indicator group of microarthropods in the study of natural landscapes and anthropogenically disturbed ecosystems [32–34]. The aim of this study was to determine the diversity of soil types, plant communities, and oribatid mites in karst relief forms under conditions of North taiga forests.

## 2. Characteristics of the Study Area

### 2.1. Physical and Geographic Conditions

Sampling was conducted in the basins of the Ukhta River and its right tributary, the Tobys River, which are located in the Ukhta District of the Komi Republic (63°19′17.7″ N, 52°52′21.5″ E; 63°20′20.1″ N, 52°54′10.2″ E; 63°25′55.9″ N, 52°58′25.3″ E) (Figure 1). The study area is located to the south of one of the Timan uplands, the Vymsko-Volskaya Ridge, in the so-called Ukhta Saddle, which is located between the Middle and Southern Timan, with heights ranging from 150 to 200 m above sea level. The Vymsko-Volskaya Ridge is represented here by a sparse chain of low hills [30].

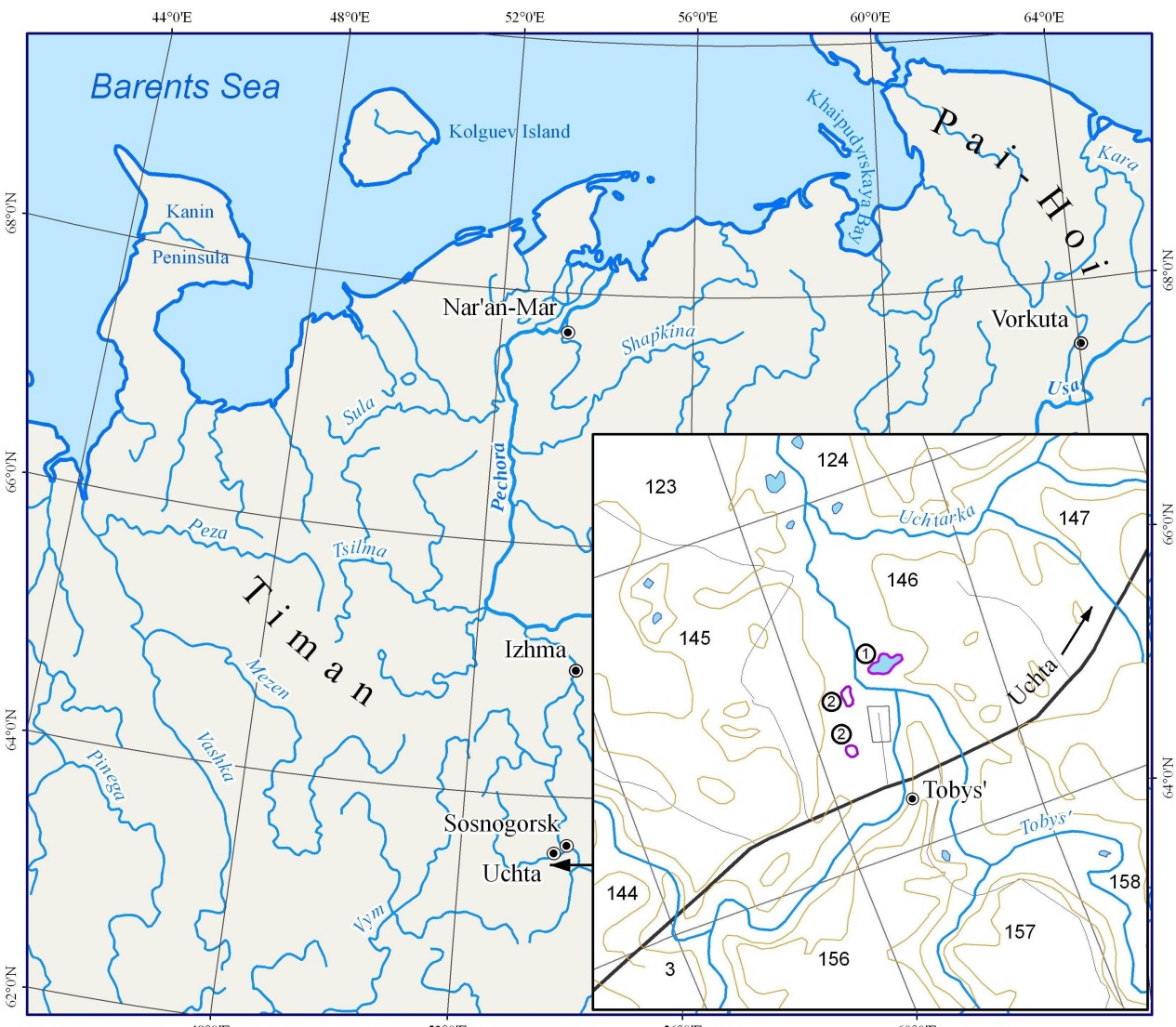

**Figure 1.** Study area in Middle Timan, Russia: 1. the Ukhta River; 2. the Tobys River.

The study area is confined to drained pine terraces, and sandy and lake plains of the northern taiga natural territorial complexes [35]. Pine forests grow mainly in conditions of good drainage, whereas spruce forests are predominant in conditions of poor drainage, and swampy areas are present in low-lying areas.

### 2.2. Climate

The study area belongs to the Atlantic–Arctic region of the temperate climate belt [36] and is characterized by an excessively humid, moderately continental climate with long winters and short, relatively warm summers [35,37]. This territory is characterized by

increased solar radiation in summer (in May–July, 10–12 kcal/cm$^2$; in August, approximately 8 kcal/cm$^2$). In winter, owing to the active invasion of cyclones that bring cloudy weather with snowfalls and blizzards, the amount of solar radiation is sharply reduced to 1–3.5 kcal/cm$^2$. A characteristic feature of the climate is frequent invasion of Arctic air masses from the Kara Sea, which is accompanied by a decrease in temperature in the summer and winter months.

The mean annual air temperature is −1.2–2.2 °C with maximum and minimum temperatures recorded as 35 °C and −49 °C, respectively. The average annual air temperature amplitude is 32–33 °C. The average annual temperature in January is −17–18 °C and in July is −15–16 °C. The duration of the frostless period is 75–95 days; with temperatures above 0 °C, it is 165–170 days; with temperatures above 5 °C (vegetation period), it is 120–125 days; and the period of active growth has temperatures above 10 °C for 75–85 days. Total annual precipitation is 600–635 mm, including 325–390 mm during the warm period (May–September), and evaporation is approximately 300–350 mm. The snow cover holds for approximately 190–200 days and its height reaches 60–70 cm. The average date of snow cover formation is October 25 to 31. The average date of snow cover destruction is 25 April to 1 May [35,38].

### 2.3. Soil-Forming Rocks

The study was conducted on the territory located among the spurs of the Timan Ridge represented by calcareous and salt-bearing rocks. In the valleys of the rivers, close to the surface, the deposits of Lower Perm can be observed, represented by terrigenous–carbonate and sulfate formations; in karst craters, this is underworn by carbonate–terrigen, red-colored deposits of the Ufa tier [39].

Pedogenesis on the eluvium of original rocks, and on the colluvial deposits are observed on the slopes of the Ukhta river terraces, in the hilly area of the Reserve. Parent rocks are represented by loams and sandy loams, including crushed stone and fragments of the underlying bedrock. Eluvium of original rocks is a weathering product of primary material. It preserves relict structural and petrographic features, genetic links, and continuous sequential transition to the original rocks. Eluvium develops on abrupt slopes (>30°) in the form of talus and pieces of original rock. Colluvial deposits are weathering products that are gravity-displaced down the slope. They accumulate at the foot and in the lower parts of the uplands, owing to the falling of fragmentary material. Colluvial deposits include the diluvium (accumulation of loose weathering products of parent rocks) in addition to boulders and rubbly boulder talus.

There are a lot of empty karst craters with depth >30 m in this territory. Most of these are not filled with water. Diluvial loams are formed at the bottom and in the lower part of the slopes of craters, as a result of the drifting and sedimentation of products of rock destruction under the effect of rain and melt waters from watersheds, and from the upper parts of the slopes of karst craters. Diluvium is layered, and commonly conspicuously graded in both horizontal and vertical directions.

Fluvioglacial deposits covering the original rock table with a thin blanket are commonly placed on the tops and upper slopes of the watershed hills, pine forest (sandy) terraces, and in the valleys of the watersheds where pine forests are widespread. Data on sand deposits including gravel and boulders are presented. Moraine boulder loans with sandy interlayers often underlie fluvioglacial deposits [40].

### 2.4. Soils

On the watershed, lowland bogs [41–44] are situated in the depressions between the glacial and sloping hills with peaty eutrophic gley silty-peat soils (Eutric Rheic Hemic Histosols). Bogs with hypnum moss or sedge cover are dominant along with suppressed trees such as spruce, birch, and pine. Soils are developed under excess moisture conditions, where significant accumulation of organic matter occurs. Peat of this soil type has medium and slightly acidic reactions.

On higher landscape positions of the watershed with a low groundwater table, iron-illuvial contact-gley podzols (Stagnic Albic Rustic Podzols (Arenic)) are formed. In the studied soils, the signs of gleying processes and accumulation of organic raw material occur owing to the presence of binomial deposits (the sandy layer is underlined by loamy strata at a depth of 42–50 cm) and bioclimatic conditions of northern taiga subzone. There are different types of soils within karst landforms. At the sites where limestones and dolomites lie at depths lower than 2–3 m of the soil surface, the area is covered by quaternary sediments of predominantly coarse texture and dense red gypsiferous strata. Thus, in the upper slope (around 30°) of the karst crater, iron-illuvial podzol (Albic Rustic Podzol (Arenic)) is formed on fluvioglacial deposits underlain by crushed stony rock at a depth of 90 cm.

In the middle slope (30–40°) of the karst craters, mucky–peaty peat-lithozem (Dystric Rendzic Folic Leptosol) is formed on weathering products (rubbly fine earth strata) of dense red gypsiferous rock due to the total drifting of the quaternary sediments mantle. The soil profile is thin, no more than 25–30 cm. In the lower part of slope and at the bottom of karst craters under additional moisture, gray-humus gleystratozem (Pantocolluvic Skeletic Stagnic Regosol (Abruptic, Ochric)) is formed.

Dense parent rocks, lying at a depth of >1 m, do not significantly affect the soil formation on lower parts of the relief because of the coverage of a mantle of fine-earth deposits that provides the conditions for a more stable leaching water regime and appearance of signs of podzolization in the soil profile. Gray-humus raw-humus podzolized residual-calcareous soil (Dystric Calcaric Skeletic Cambisol (Ochric, Nechic)) is located in the Ukhta River valley, on the upper gentle slope (5–7°).

Soil is formed on quaternary sediments, relatively near the surface (within 1 m), underlying a dense carbonate rock table. Soil profile thickness is 50–60 cm, there is a podzolized, depleted AYe-horizon. At the same place, in the Ukhta river valley on the lower steep slope (around 30–35°) of the carbonate rocks outcrops, mucky–dark-humus carbolithozem (Calcaric Mollic Folic Leptosol) is formed. Carbolithozems are specific soils with thin profiles, no more than 30–40 cm in depth.

### 2.5. Vegetation

According to the geobotanical zoning of the Non-Black Earth Region of the European part of the Russian Federation (1989), the territory of the Middle Timan is located in the strip of the northern taiga forests of the Vychegodsko-Pechorskaya subprovince of the North European taiga province [45]. According to the geobotanical zoning adopted in the Komi Republic, the upland part of the Middle Timan belongs to the Middle Timan district of spruce, pine, and birch forests, in which spruce and larch–spruce green-moss forests dominate; in stream valleys and floodplains, large, grass–spruce forests have developed [38].

*Picea obovata* Ledeb. is the dominant forest-forming species. Green-moss spruce forests, in which *Betula pubescens* Ehrh. is common, prevail in drained areas of the watersheds, and spruce forests with an admixture of *Pinus sylvestris* L.—belonging to long-moss and sphagnum forest types—are present in wetland interfluves. Permafrost terraces and fluvioglacial plains are covered with licheniferous and green moss pine forests. A distinctive feature of the forests of the Middle Timan is the broad distribution of *Larix sibirica* Ledeb., which, in these conditions, forms the largest massifs of larch forests in the Komi Republic or occurs as an admixture. The main associations of larch forests in the Middle Timan are lingonberry and bilberry–green-moss, fern, large grass, and shrubby grass forests; pine–larch and larch–birch forests are also present [38].

### 3. Material Collection and Processing

#### 3.1. Soils

Soil chemical analyses were performed at the certified Ecoanalytical Laboratory of the Institute of Biology (Komi Science Center, Urals Branch of the Russian Academy of Sciences) (certificate ROSS RU.0001.511257 from September 2019). The pH value was

determined potentiometrically using the HANNA instrument HI 8519N (Portugal) at a soil:solution ratio of 1:25 and 1:5 for organic and mineral horizons, respectively [46]. Hydrolytic (total) acidity was determined by the Kappen method. Exchangeable cations ($Ca^{2+}$ and $Mg^{2+}$) content was determined using the method described by Gedroits with 1M $NH_4Cl$ solution and subsequent determination of desorbed cations on an ICP Spectro Ciros CCD (Germany) device. Concentrations of mobile forms of phosphorus and potassium were determined in 0.2 N HCl-extract from soil according to Kirsanov, and the carbonate content was estimated by the volumetric method [47]. The content of total carbon ($C_{tot}$) was estimated by the gas chromatography method on an elemental CHNS-O analyzer (EA 1100; Carlo Erba, Italy). Organic carbon content ($C_{org}$) was considered to be equal to $C_{tot}$ because parent rocks ($pH_{H2O} < 6.5$) contained inorganic carbon ($C_{inorg}$) in trace concentrations and these concentrations did not exceed the method's error. The calculation of reservoir levels of elements (Q) in particular soil horizons (layers) was performed by multiplying soil bulk density (g cm$^{-3}$), thickness of horizon (cm), and content of the corresponding $C_{org}$ element, % [48]. The total reserves of elements were calculated by a simple summation $Q\Sigma = Q1 + Q2 + \ldots + Qn$, where n = number of horizons (layers). The names of soils and diagnostic horizons were identified following the Russian Soil Classification System [49] and the World Reference Base for soil resources [50]. Color by the Munsell system (Standard Soil Color Charts) is given for air-dried soil samples.

The spatial heterogeneity of the object of study was assessed by laying the main soil section and excavations including organic horizons. The convergence and reproducibility of the results was controlled by repeated determinations. Each soil sample was analyzed in triplicate; the quality control of the determination was carried out using control samples on calibrated equipment in an ecoanalytical laboratory. The statistical processing of the results was performed using the Statistica and MS Excel software packages. The arithmetic mean values of the studied parameters and standard deviation were calculated. Dendrograms of the similarity of horizons and soil sections were constructed using the weighted average method. The Euclidean distance was used as a measure of difference [51].

### 3.2. Plant Communities

Forest phytocenoses were described based on 400 m$^2$ sampling plots using standard methods accepted in geobotany and forest typology [52,53]. Bog vegetation communities were described based on 25 m$^2$ sampling plots. Species comprising the herbaceous–shrub layer and moss–lichen cover were evaluated as a percentage of the species' projective cover from 100% of the total projective cover of the sample area. Species composition lists of local flora were documented using herbarium collections stored in the herbarium of the Komi Biology Institute, Ural Branch of the Russian Academy of Sciences (SYKO). The plants were identified using the monograph Flora of the North-East of the European part of the USSR [54–57]. Plant names are listed as per the Plants of the World Online database (http://www.plantsoftheworldonline.org/) (accessed on 1 January 2022) [58].

### 3.3. Soil Oribatid Mites

Sampling for soil microarthropods was conducted in seven plant communities: sedge lowland bog, blueberry–green moss pine forest, dwarf-lichen–green moss mixed forest, herbaceous–green moss mixed forest, green moss–herbal spruce forest, dwarf–herbaceous postfire mixed forest and herbaceous thin mixed forest (Sites 1–7). Invertebrate specimens were collected according to standard methods [32]. Twelve soil samples, which included dead litter, each with dimensions of 5 × 5 cm by 10 cm-deep, were collected from each of the sites in July 2020. The distance between the selection points in one series of samples was 7–8 m. A total of 84 soil samples were collected. The soil samples were placed in plastic bags and then in boxes from cardboard to avoid deformation of the soil monolith; 2 days after the selection, they were transported by rail to the Institute of Biology of the Komi Scientific Center (IB Komi SC UB RAS), Syktyvkar, where they were immediately placed into Tullgren soil extractors. The microarthropod fauna was extracted under 40-Watt

bulbs into 96% alcohol over a period of 7–10 days until the soil was completely dry. The Oribatida (Figure 2) were identified to the species level using morphological taxonomic characters [59]. Identification of oribatid mites was performed by one of the authors, namely, E.N. Melekhina. A total of 10,200 specimens of adult oribatid mites were identified up to the species level. Biological material is deposited at the Institute of Biology of the Komi Scientific Center (Syktyvkar, Russia).

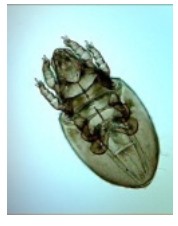

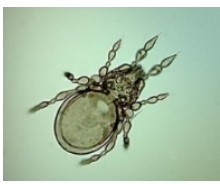

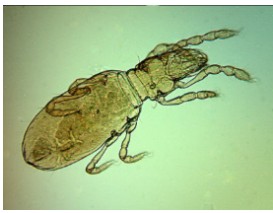

*Malaconothrus (M.)*
*monodactylus*
(Michael, 1888)

*Suctobelbella* sp.

*Eulohmannia ribagai*
(Berlese, 1910)

**Figure 2.** Some of oribatid mites taxa recorded in Middle Timan, Russia (Foto E.N. Melekhina).

Taxonomies of oribatid mites and types of global distribution of the species follow Subías [60]. The classification of life forms of oribatid mites is given according to D.A. Krivolutsky [32]. The collections included varying species: inhabitants of the soil surface and upper horizons of the litter (epigeic), inhabitants of the litter layer (hemiedaphic), inhabitants of small soil holes (euedaphic), eurybionts and hydrobionts, as well as non-specialized species. Diversity indices of oribatid mites, the Shannon index and the Berger–Parker index, were calculated in the program PAST V 3.0 [61]. The reliability (significance) of differences in the population density of oribatid mites was determined using the Mann–Whitney U-test at a significance level of $p < 0.05$; PAST V 3.0 software was used for data processing [61].

## 4. Results and Discussion

### 4.1. Description of Soil

In the karst landscapes of the Middle Timan, the soil cover was highly mosaic and distinct. In sedge lowland bogs (Site 1), peaty eutrophic gley silty-peat soil (Eutric, Rheic, Hemic Histosol) was common. The soil profile was of the TE–TEmr–Ghi combination. The peat horizon was separated into two subhorizons. The upper TE-subhorizon was 10–20 cm-thick; brownish (2.5Y 4/2); and contained decomposed residues of mosses, sedge, and litters of leaves, needles, and other residues of woody vegetation. The lower peat layer (TEmr-horizon) was moist, silty, and characterized by a high degree of decomposition (up to 40–50%) of organic material with dark (5Y 2/2) to black, (5Y 2/1) color. A dirty-bluish (5Y 5/2) Ghi-horizon lay beneath the peat layer and above the underlying rubble gritty–sandy eluvium. Salt extract pH varied from 4.6 to 5.0 (Table 1). The presence of less acidic peat below a depth of 20 cm appears to be due to the effect of a high-level mineralized groundwater table. Hydrolytic acidity in the upper horizon exhibited high values that decreased downward in the profile. The upper peat horizon likely begins to separate from the mineral rich groundwater supply, and starts to feed only upon precipitation (e.g., rainfall, snow). A high content of mobile forms of phosphorus and potassium was present only in the upper horizons of eutrophic peat soil and decreased downward in the profile.

**Table 1.** Chemical composition of soils.

| Horizon | Depth, cm | pH ($_{H2O}$) | pH ($_{KCl}$) | $C_{tot}$ ($C_{inorg}$) | Exchangeable Cations | | $Ac_{tot}$ * | Mobile forms | | ** V, % |
|---|---|---|---|---|---|---|---|---|---|---|
| | | | | | $Sa^{2+}$ | $Mg^{2+}$ | | $P_2O_5$ | $K_2O$ | |
| | | | | | cmol(+)/kg | | | mg/kg | | |
| *Peaty eutrophic gley silt-peat soil (Eutric Rheic Hemic Histosol)* | | | | | | | | | | |
| TE | 0–12 | 7.20 | 6.82 | - | 42.3 | 26.2 | 75.7 | 240 | 227 | 47.5 |
| TEmr | 12–40 | 7.02 | 6.54 | - | 48.5 | 14.1 | 26.5 | 50 | - | 70.3 |
| Ghi | 40–46 | 6.84 | 6.10 | - | 43.4 | 9.1 | 21.3 | 50 | 86 | 71.1 |
| *Iron-illuvial contact-gley podzol (Stagnic Albic Rustic Podzol (Arenic))* | | | | | | | | | | |
| O | 0–6 | 5.95 | 4.21 | - | 22.5 | 8.0 | 50.3 | 8870 | 22,930 | 30.4 |
| E | 6–17 | 4.45 | 3.42 | - | 1.0 | 0.5 | 5.9 | 530 | 690 | 1.5 |
| EBF | 17–20 (32) | 5.11 | 4.34 | - | 0.2 | 0.2 | 3.8 | 690 | 420 | 0.4 |
| BFg | 20 (32)–42 | 5.02 | 4.26 | - | 0.3 | 0.2 | 4.0 | 470 | 360 | 0.5 |
| Dg | 42–62 | 5.25 | 3.78 | - | 3.3 | 1.7 | 8.6 | 270 | 1230 | 4.9 |
| *Iron-illuvial podzol (Albic Rustic Podzol (Arenic))* | | | | | | | | | | |
| O | 0–3 (4) | 4.31 | 3.46 | 41.4 | 13.3 | 3.5 | 53.8 | 2850 | 11,360 | 16.8 |
| E | 3 (4)–20 | 5.44 | 4.50 | 0.75 | 0.4 | 0.1 | 2.7 | 540 | 250 | 0.5 |
| BF | 20–56 | 5.41 | 4.22 | 0.54 | 1.1 | 0.3 | 5.2 | 530 | 530 | 1.4 |
| B | 56–92 | 5.59 | 3.92 | 0.09 | 3.9 | 1.8 | 6.4 | 70 | 980 | 5.7 |
| BC | 92–120 | 5.50 | 4.01 | - | 4.9 | 2.0 | 3.0 | 160 | 870 | 6.9 |
| Dg | 120–130 | 5.61 | 4.40 | - | 3.6 | 1.5 | 3.5 | 160 | 790 | 5.0 |
| *Mucky–peaty peat-lithozem (Dystric Rendzic Folic Leptosol)* | | | | | | | | | | |
| T1 | 0–3 | 5.00 | 4.11 | 40.9 | 33.2 | 7.6 | 53.8 | 4260 | 13,200 | 40.8 |
| T2 | 3–10 | 4.55 | 3.55 | 32.0 | 29.0 | 5.4 | 65.9 | 2490 | 9440 | 34.5 |
| TH | 10–14 | 6.10 | 5.41 | - | 64.4 | 4.2 | 32.8 | 3440 | 7190 | 68.6 |
| (C) | 16–28 | 7.22 | 6.49 | 12.9 (0.71) | 12.2 | 0.7 | 0.8 | 170 | 610 | 12.9 |
| R | 28–58 | 7.59 | 6.82 | - | 11.3 | 1.0 | 0.6 | 240 | 690 | 12.2 |
| *Gleyic gray-humus stratozem (Pantocolluvic Skeletic Stagnic Regosol (Abruptic, Ochric))* | | | | | | | | | | |
| AYao | 0–2 | 6.26 | 5.51 | 9.50 | 54.6 | 8.3 | 28.7 | 5940 | 9690 | 63.0 |
| AY | 2–10 | 6.01 | 5.30 | 2.10 | 14.7 | 2.7 | 6.4 | 1210 | 1450 | 17.5 |
| RY1g | 10–22 (26) | 6.28 | 5.34 | 0.98 | 14.0 | 2.5 | 4.7 | 1290 | 1150 | 16.5 |
| RY2g | 22 (26)–54 | 6.70 | 5.51 | 0.57 (0.10) | 4.9 | 1.2 | 1.4 | 2990 | 660 | 6.2 |
| *Gray-humus raw-humus podzolized residual-calcareous soil (Dystric, Calcaric, Skeletic Cambisol (Ochric, Nechic))* | | | | | | | | | | |
| O | 0–2 (3) | 5.98 | 5.82 | - | 46.6 | 14.2 | 27.4 | 14,660 | 31,930 | 60.8 |
| AYao | 2 (3)–10 (15) | 5.81 | 5.13 | 25.40 | 19.0 | 7.6 | 29.3 | 8990 | 19,340 | 26.6 |
| AYe | 10 (15)–23 | 5.05 | 3.96 | 1.19 | 0.9 | 0.6 | 5.9 | 850 | 740 | 1.6 |
| AC1 | 23–32 | 5.46 | 3.95 | - | 5.7 | 2.2 | 7.4 | 780 | 1560 | 7.9 |
| AC2 | 32–54 | 5.74 | 4.05 | - | 7.0 | 2.5 | 4.1 | 820 | 1290 | 9.5 |
| Cca | 54–79 | 6.56 | 5.58 | - | 10.0 | 3.3 | 2.3 | 970 | 1400 | 13.3 |
| Rca | 79–85 | 7.50 | 6.78 | - | 16.2 | 4.0 | 0.8 | 30 | 1170 | 20.1 |
| *Mucky–dark-humus carbolithozem (Calcaric Mollic Folic Leptosol)* | | | | | | | | | | |
| AH1 | 0–7 | 6.56 | 6.27 | 5.9 (1.0) | 59.8 | 19.7 | 21.9 | 7980 | 14,900 | 79.4 |
| AH2 | 7–12 | 7.38 | 7.07 | 5.0 (0.68) | 49.4 | 16.4 | 17.5 | 21,860 | 4830 | 65.8 |
| Bmh | 12–20 | 7.65 | 7.59 | 2.8 (0.42) | 40.4 | 13.6 | 8.8 | 13140 | 2000 | 54.0 |
| (Cca) | 20–30 (31) | 7.84 | 7.85 | - | 4.6 | 1.5 | 0.3 | 320 | 220 | 6.1 |
| Rca | 30 (31)–50 | 7.86 | 7.84 | - | 4.2 | 1.3 | 0.2 | 300 | 140 | 5.5 |

Note. Dashes indicate the absence of determination. * Total (hydrolytic) acidity. ** The degree of saturation with bases shows the percentage of the total amount of base cations (S) in the soil absorption complex from the total absorption capacity (T) and is expressed by the formula V = (S/T) × 100.

On the upper landscape positions (Site 2), iron-illuvial contact-gley podzol (Stagnic Albic Rustic Podzol (Arenic)) was formed under blueberry–green moss pine forests. The soil profile exhibited the O–E–BFg–Dg combination (Figure S1). These soils are represented by expressed profile differentiation into genetic horizons and small thickness of the podzolic horizon (around 10 cm); moreover, the E-horizon is heterogeneous and partly discontinuous. In the study area (transition zone between the middle and northern taiga) under current climatic conditions, the shallowness of the E-horizon is a specific feature of podzols. Dark-brownish (7.5YR 2/3) forest litter (thickness 6–8 cm), including numerous roots, was above the whitish (7.5YR 6/1) sandy podzolic horizon underlined by the gley Al-Fe-humus horizon, and characterized by the thick humus–ferruginous coatings on faces of mineral grains and bridgelike connecting sand particles. The BFg-horizon was yellow–ocher (10YR 6/4); in addition, bluish toning, ocher–rusty spots, concretions, and smears were present. Boulders and pebbles with thin and semiskeletal roots were present throughout the profile. Stoniness was approximately 35% in the BFg-horizon. At the depth of 42 cm, the soil profile contained medium loam layer with rare sand lenses (Dg-horizon), characterized by cloddy, finely subangular–blocky structure, brownish–grayish brown (10YR 7/3) in color, with the morphochromatic signs of gleying by way of bluish and small rusty spots. Concretion neoformations were present with the presence of thin roots. In the Dg-horizon, stoniness was less than 10%. Chemically, iron-illuvial podzols in their natural state are of low fertility. They are characterized by relatively high acidity and very low content of mobile forms of phosphorus and potassium (Table 1). The specific feature of podzols formed on binomial deposits is the distribution of exchangeable cations in the soil profile; large quantities are observed in the upper litter-peat O-horizon, cations are basically absent in the lower lying (EBF and BFg) horizons, and only in the underlying loamy horizon does their content slightly increase again. In the podzolic E-horizon, the relatively high $C_{tot}$ content is associated with both its removal from the upper litter layer and the accumulation of weakly humified plant residues in this horizon.

Within karst landforms, on the upper slope (around 30°) of the karst crater (Site 3), iron-illuvial podzol (Albic Rustic Podzol (Arenic)) is formed under dwarf-lichen–green moss mixed forests. The soil profile is of the O–E–BF–BC–Dg–R combination (Figure S1). The underlying yellow–brownish (7.5YR 3/2) forest floor was of 1.8–3 cm-depth with black organogenic inclusions and the presence of numerous roots.

The podzolic E-horizon is placed and characterized by sandy texture, whitish (7.5YR 6/2) in color, with black and dark-gray tongues and interlayers in the upper part, and small-size coaly particles and numerous roots of different diameters. The thickness of this horizon varied widely (5 to 16 cm). On the higher landscape positions, the E-horizon was basically sporadic and heterogeneous. Under the E-horizon, the lower lying sandy ocher (7.5YR5/6) BF-horizon with many roots, changed into the yellowish-pale-brownish (7.5YR 5/4) BC-horizon with similar textural content. The lower profile part (70–90 cm-depth) was brownish (10YR 5/6), sandy loam, sandy, subangular blocky structured, with rusty spots and stains, dark ocher smears, and small-sized concretion neoformations. Chemically, podzols were characterized, as noted in the description of the soil at Point 2, by a small amount of exchangeable bases; relatively high acidity (Table 1); and as noted previously upon description of Stagnic Albic Rustic Podzol (Arenic). The high content of $C_{tot}$ indicates the raw-humus nature of the organic matter. The fixation of humic acids into the mineral BF-horizon was owing to the formation of humic–ferruginous complexes. In the study soil series, the organic carbon stocks were minimal (8.3 kg m$^{-2}$) in podzols (Figure 3). The soils of this bioclimatic zone are characterized by the high mobility of humus compounds and humus infiltration, which determine the relatively high $C_{org}$ content in the mineral part of the profile, especially in the litter horizons (up to 1.0–1.4%). Therefore, this was reflected in a more significant redistribution of $C_{org}$ stocks in the podzol profile—the share of organic horizons was approximately 65.4% of the total organic carbon stocks. In the podzol profile, humic substances are basically represented by highly mobile components extracted with 0.1 N NaOH solution. In the soil profile, the share of these was approximately 34.1% of

the total stocks of $C_{org}$, which was 1.9–2.5 times higher than that in the soils formed on calcareous and residual-calcareous deposits.

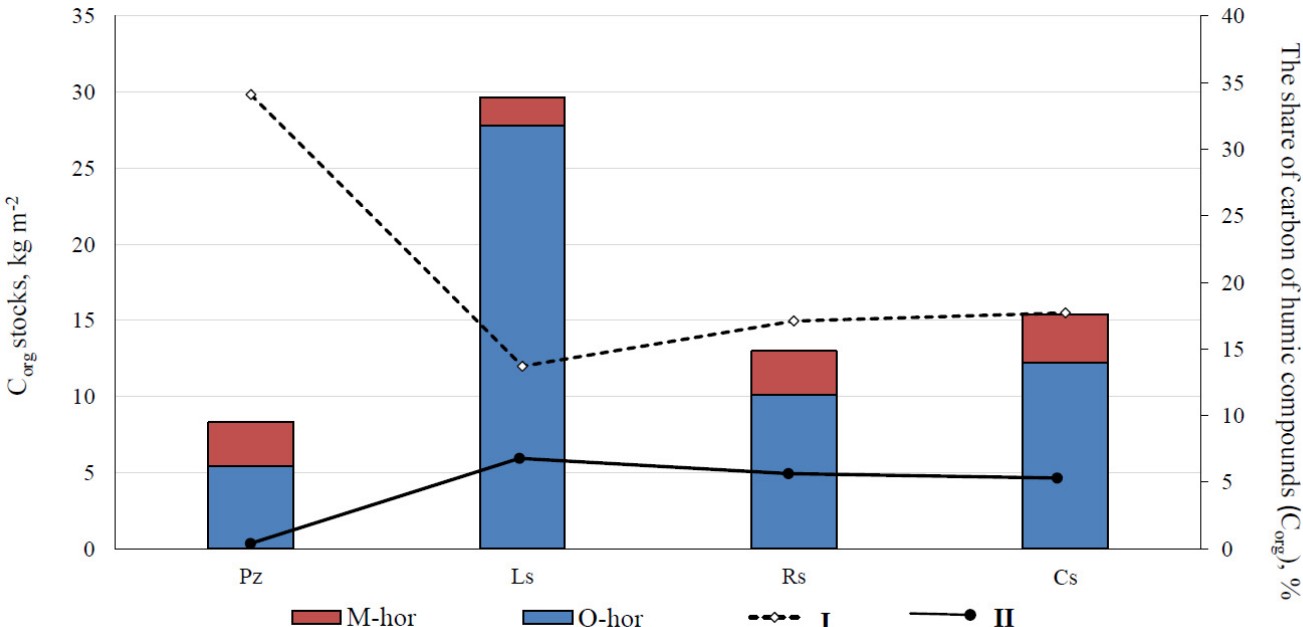

**Figure 3.** $C_{org}$ stocks in the organogenic (O-hor) and mineral (M-hor) soil horizons and the share of mobile humic substances extracted by 0.1 N NaOH (I) solutions in carbon stocks and humic substances fixed in the soil as complexes with calcium and magnesium (II). Soils: Pz, Iron-illuvial podzol (Albic Rustic Podzol (Arenic)); Ls, Mucky–dark-humus carbolithozem (Calcaric Mollic Folic Leptosol); Rs, Gleyic gray-humus stratozem (Pantocolluvic Skeletic Stagnic Regosol (Abruptic, Ochric)); Cs, Gray-humus raw-humus podzolized residual-calcareous soil (Dystric Calcaric Skeletic Cambisol (Ochric, Nechic)).

In the middle slope (30–40°) of the karst crater, Mucky–peaty peat-lithozem (Dystric Rendzic Folic Leptosol) was formed under herbaceous–green-moss mixed forests (Figure S1). The soil profile presented the TH–(C)–R combination (Site 4). The soil was characterized by peat Th-horizon up to 14 cm-thick, comprising a homogeneous mixture of organic material with different decomposition degrees, with numerous white and yellow fungal hyphae and lots of roots, and an uneven color (10YR 2/3). The lower part of the peat layer was commonly musky, lard, and black (10YR 2/2), with dark-brownish incls. The yellowish–dark brownish (10YR 4/4) interlayer of fine-earth material painted by organic matter was often located between large slabs of brick-brownish rocks and the peat horizon. The horizon contained small fragments of underlying stone rocks. In this soil, the organogenic horizons are characterized by high hydrolytic acidity, acidic and slightly acidic reaction of the water extract, and a $pH_{HCl}$ range from 3.6 to 5.4 (Table 1). In the fine-earth layer, the pH values were neutral. The organogenic horizons contain the maximum amount of exchangeable bases and mobile forms of potassium and phosphorus in comparison with the fine-earth layer, where their content sharply decreases. Their low content is noticeable, opposed to that of the Calcaric Mollic Folic Leptosols, formed on the slopes, but with a close underlying carbonate rock table. This is because of the low ash content of the litter and intensive leaching of bases from residues under conditions of low temperature and leaching water regime.

In the lower slope and at the bottom of the karst craters with the additional moisture, gleyic gray-humus stratozem (Pantocolluvic Skeletic Stagnic Regosol (Abruptic, Ochric)) (AYao–RYg–D) was formed under green-moss–herbal spruce forests (Site 5) (Figure S1). The stratozem profile was layered and heterogeneous, consisting of several gray-humus and stratified subhorizons of different composition and structure, lying on the fine-earth–rubble (pebble) layer or directly on the dense bedrock. This soil was characterized by a

brownish–gray humus-accumulative layer 22–26 cm-thick. The layer was dark and bright (10YR 2/2) at the upper border and changed to light and pale (10YR 4/2, 10YR 4/3) downward the profile. It had an admixture of weakly decomposed plant residues, sandy loam texture, with a small amount of bleached gruss and sand grains, and large brick-brownish pieces of parent rock. The lower-lying sandy RYg-subhorizons were represented by a brownish color (10YR 4/2) with morphochromatic signs of gleying (rusty-ocher and brownish spots, streaks and smears, small dense ferruginous nodules, and concretion neo-formations) and small-sized rubble. In the subhorizons, stoniness was 20% and increased downward the profile together with an increasing degree of profile gleization. The sand parent rock (D-horizon) underlaid the soil layer at a depth of 117–125 cm. Stratozems are characterized by heterogeneous texture. The ratio of physical sand to clay is 3:1 [62]. A weak degree of podzolization appeared in increasing acidity of the upper horizons and in the redistribution of exchangeable bases, and the absorption complex was unsaturated with bases (Table 1). $C_{org}$ stock in the stratozem profile was 13.0 kg m$^{-2}$ (Figure 3). The relatively low reserves of $C_{org}$ are owing to the lower thickness of the soil profile (up to 55 cm) and relatively low $C_{org}$ content both in the organic horizon and in the mineral part of the profile. The composition of soil organic matter and contribution of weakly decomposed plant debris, mobile humic substances, and humic substances associated with exchange bases into the $C_{org}$ stocks corresponded to 78.0%, 17.1%, and 4.6%, respectively.

In the Ukhta river valley, on the upper gentle slope (5–7°), gray-humus raw-humus podzolized residual-calcareous soil (Dystric Calcaric Skeletic Cambisol (Ochric, Nechic)) was formed under dwarf-herbaceous postfire mixed forests. The soil profile had the O–AYe–AC–Cca–Rca combination (Site 6) (Figure S1). Gray-humus soils are formed under automorphic conditions and have a leaching water regime. The accumulative part of the profile was represented by yellowish–grayish–brown (10YR 3/2) forest litter (O-horizon), containing leaf litter, twigs, and needles, with numerous roots and black organogenic inclusions. The degree of litter decomposition increased with depth. The humus Aye-horizon was gray with brownish toning (10YR 2/2), and characterized by signs of podzolization (light color, bleached grains) in the lower part and numerous incls of small black smears and coal-like particles. There were several roots in this horizon. In the evaluated soil, pyrogenic signs were observed, which were preserved for a long time after fire occurrence. The mineral component of the profile was weakly differentiated. The gray-humus horizon transited slowly into the parent rock. The Cca-horizon was sandy loam, yellowish–dirty gray (10YR 6/4); gray toning was due to humus infiltration. The horizon contains a fine-grained rubble and powdery fine earth; thin roots are present. There were fragments of carbonate rock with the amount and size increasing downward the profile. The gray-humus soil profile was underlined by dense carbonate rock at a depth of 54–60 cm. The soil exhibited effervescence in the presence of 10% HCl solution. $C_{org}$ content was 1.2% in the mineral layer and up to 25.4% in the organogenic horizon (Table 1). In the Cambisol profile, total stocks of Corg were approximately 15.4 kg m$^{-2}$ (Figure 3). The total $C_{org}$ stocks (79.4%) were located in the organogenic part of the profile; however, the more significant humus content of the subsurface mineral horizons determines their redistribution in the profile. The contribution of weakly decomposed plant residues into the soil carbon content was still at a high level (78.3%). However, in this soil type, there is an increasing share of specific components of humus in the composition of soil organic matter: both mobile and extractable 0.1N NaOH, forming complexes with exchange bases—17.1% and 4.6% of the total $C_{org}$ stocks, respectively. The obtained data on the Dystric Calcaric Skeletic Cambisol (Ochric, Nechic) were close to the corresponding indicators of the Pantocolluvic Skeletic Stagnic Regosol (Abruptic, Ochric). This indicates the similarity of the conditions of humus formation in the studied soils.

In the Ukhta river valley, on the lower steep slope of approximately 30° (Site 7), mucky–dark-humus carbolithozem (Calcaric Mollic Folic Leptosol) was formed under herbaceous thin mixed forests (Figure S1). The soil profile had the AH–(Cca)–Rca combination. Owing to the thin forest stand, the illumination of the soil surface and the projective

cover of herbaceous plants increased, and soil is supplied with mineralized slope water. In these soils, the formation of lichen–moss litter and rhizogenous humus, saturated with calcium, and a soddy accumulative process occurs. As a result, a mucky–dark-humus horizon of dark gray (10YR 2/1) to black (10YR 1.7/1) color is formed, characterized by mechanical mixtures of different degrees of residue decomposition with mineral components; granular structure, penetrated by numerous roots; and abrupt boundary with the underlying Csa-Csa-horizons (10YR 8/1 and 10YR 7/3). The yellowish–brownish (10YR 6/2) fine-earth–gruss mass was under the AH-horizon, below brownish-yellow (10YR 7/2); this was formed owing to the in situ transformation of dense carbonate rocks into the fine earth. The soil profile was underlined by massive slabs. Soil matrix showed violent effervescence in the presence of 10% HCl solution for both fine earth and rubble of carbonate parent rock.

In this case, the genesis of the upper horizons of carbolithozems (up to 15 cm-thick) is determined by the specifics of soil-forming conditions: high rubble content of lower layers, sandy loam texture of the upper layers, and humification processes occurring upon conditions of high carbonate content and migration of organic compounds downward the profile due to the leaching water regime. These factors together with the dominated calciphytic vegetation with reference to the specifics of pigmentation, conservation, decomposition, and biological activity of microflora, lead to dark coloring of the upper layer of the study soils [22,63]. These soils have a neutral and slightly alkaline environment, owing to the predominance of $Ca(HCO_3)_2$ in the soil solution, a low hydrolytic acidity value, and significant amount of calcium and magnesium in the content of exchangeable cations, which allows for classification of these soils as saturated with bases (V = 78–96%) (Table 1). Such chemical indicators are not typical for the soils of the northern part of the middle taiga subzone. Humification processes are primarily affected by the general bioclimatic conditions of the taiga zone: low biochemical activity and short period [63,64]. The total stocks of soil organic carbon are approximately 30 kg m$^{-2}$ in the profile of such soils; almost 93–94% of $C_{org}$ stocks accumulate in the organogenic horizons (Figure 3). In the mineral layers, the share of $C_{org}$ stocks is less than 2%. The main part of the soil organic matter (approximately 80% of the total $C_{org}$ stocks) was represented by weakly decomposed plant residues. The share of mobile humus components extracted in 0.1N NaOH was 13.7% of the total $C_{org}$ stocks in the soil. The presence of a significant amount of exchange bases in the soil (Table 1) promotes the partial fixation of humus substances of insoluble calcium and magnesium salts in the profile of carbolithozems. The share of such humus components was up to 5.6% of the soil carbon stocks (Figure 3).

Soil cover exhibited high mosaicism and uniqueness. Acidic and weakly acidic soil reactions, high hydrolytic acidity values, and insignificant content of exchangeable bases were typical for Podzols that formed on automorphic relief positions. Rubbly, incompletely developed thin soils were formed on the slopes of karst crater and ridges with the organogenic horizons distinguished by relatively high acidity, high exchangeable base content, and mobile forms of potassium and phosphorus.

### 4.2. Plant Communities

The vegetation cover of the studied area was dominated by forests both in the river valleys and watersheds (Table 2; Figure S2). Watersheds contained relatively small areas of bogs, predominantly transitional. Floodplain meadows and shrub thickets formed by *Salix dasyclados* with an admixture of *Duschekia fruticosa* are common in the Tobys and Ukhta river valleys.

**Table 2.** Species composition of plant communities at sampling points.

| Sampling Sites | Plant Community, Plant Species | Soil Type |
|---|---|---|
| | Depression between glacial hills and uplands | |
| Site 1 | Sedge wetland community in a runoff trough<br>63°19′17.7″ N, 52°52′21.5″ E, 153 m asl | Peaty eutrophic gley silt–peat soil (Eutric Rheic Hemic Histosol). Soil profile has the TE–TEmr–Ghi combination. |
| | Grass layer. GPC 40–50%. Woollyfruit sedge (*Carex lasiocarpa* Ehrh.) PC 10–15%, beak sedge (*Carex rostrata* Stokes) PC 10–15%, blister sedge (*Carex vesicaria* L.) PC 15–20%, globular-spike sedge (*Carex globularis* L.), thread rush (*Juncus filiformis* L.). | |
| Site 2 | Pine–bilberry–green-moss forest. Stand formula Pinus 7 Betula 2 Larix 1 Picea<sup>+</sup> Populus<sup>+</sup>.<br>63°19′22.5″ N, 52°52′24.3″ E, 155 m asl | Iron-illuvial contact-gley podzol (Stagnic Albic Rustic Podzol (Arenic)). Soil profile has the O–E–BFg–Dg combination. |
| | Tree layer. Common pine (*Pinus sylvestris*) 17–19 m, downy birch (*Betula pubescens*) 15–17 m, Siberian larch (*Larix sibirica*) 20–22 m, Siberian spruce (*Picea obovata*) 15–17 m, trembling poplar or aspen (*Populus tremula* L.) 15–17 m.<br>Undergrowth. The birch (*Betula pubescens*).<br>Undergrowth. common juniper (*Juniperus communis* L.) and prickly wild rose (*Rosa acicularis* Lindl.) PC 5–10%.<br>Grass–bush layer. GPC 70–80%.<br>Shrubs. Bilberry (*Vaccinium myrtillus* L.) PC 60%, lingonberry (*Vaccinium vitis-idaea* L.).<br>Grasses. Twisted pike (*Avenella flexuosa* (L.) Drejer), wood horsetail (*Equisetum sylvaticum* L.), annual sycamore (*Lycopodium annotinum* L.), northern lignea (*Linnaea borealis* L.), wood geranium (*Geranium sylvaticum* L.), common golden rose (*Solidago virgaurea* L.), narrow-leaved willow-herb (*Epilobium angustifolium* L.), two-leaved mayonnaise (*Maianthemum bifolium* (L.) F.W. Schmidt.), Sedmichia europaea (*Lysimachia europaea* (L.) U. Manns and Anderb.).<br>Moss cover. GPC 30–40%. Green mosses *Hylocomium splendens* (Hedw.) Bruch et al., *Pleurozium schreberi* (Brid.) Mitt. | |
| | karst crater | |
| | Upper part of the karst crater slope | |
| Site 3 | Mixed motley grass forest. Forest stand formula Pinus 3 Picea 2 Betula 2 Populus 3 Larix<sup>+</sup>.<br>63°20′20.1″ N, 52°54′10.2″ E, 163 m asl | Iron-illuvial podzol (Albic Rustic Podzol (Arenic)). Soil profile has the O–E–BF–BC–Dg–R combination. |
| | Tree layer. Common pine (*Pinus sylvestris*) 17–19 m, Siberian spruce (*Picea obovata*) 17–19 m, downy birch (*Betula pubescens*) 15–17 m, trembling poplar or aspen (*Populus tremula*) 15–17 m, Siberian larch (*Larix sibirica*) 13–15 m, goat willow (*Salix caprea* L.) 5–7 m.<br>Undergrowth. common juniper *Juniperus communis*, *Daphne mezereum* L.<br>Grass–bush layer. GPC 35–40%.<br>Shrubs: lingonberry (*Vaccinium vitis-idaea*).<br>Herbs. * Yellow lady's slipper (*Cypripedium calceolus* L.), * fragrant orchid (*Gymnadenia conopsea* (L.) R. Br.), * dremlik dark red (*Epipactis atrorubens* (Hoffm.) Besser, wood geranium (*Geranium sylvaticum* L.), wild strawberry (*Fragaria vesca* L.), wild angelica (*Angelica sylvestris* L.), spring vetchling (Lathyrus *vernus* (L.) Bernh.), meadow bluegrass (*Poa pratensis* L.), sheep fescue (*Festuca ovina* L.), palm sedge (*Carex digitata* L.), forest hawkweed (*Hieracium altipes* H. Lindb. ex Dahlst.), forest pea (*Vicia sylvatica* L.), wilted pearl pea (*Melica nutans* L.), dog violet (*Viola canina* L.), green polophyllum (*Dactylorhiza viridis* (L.) R. M. Bateman, Pridgeon, and M. W. Chase), * histote bitterling (*Polygala amarella*).<br>Moss cover. GPC 50–60%. Green mosses: *Hylocomium splendens*, *Pleurozium schreberi*. | |

**Table 2.** *Cont.*

| Sampling Sites | Plant Community, Plant Species | Soil Type |
|---|---|---|
| | Middle part of the karst crater slope | |
| Site 4 | Mixed herb–grass–strawberry forest. Forest stand formula Pinus 1 Picea 2 Betula 5 Populus 2. 63°20′20.1″ N, 52°54′10.2″ E, 155 m asl | Mucky–peaty peat-lithozem (Dystric Rendzic Folic Leptosol). Soil profile has the TH– (C)–R combination. |
| | Tree layer. Common pine (*Pinus sylvestris*) 17–19 m, Siberian spruce (*Picea obovata*) 17–19 m, downy birch (*Betula pubescens*) 15–17 m, trembling poplar or aspen (*Populus tremula*) 15–17 m. Undergrowth. Birch (*Betula pubescens)* 3–5 m and Siberian spruce (*Picea obovata*) 2–4 m. Undergrowth. Prickly wild rose (*Rosa acicularis*) up to 1 m. Grass layer. GPC 55–60%. Herbs. Basilica small (*Thalictrum minus* L.), * dremlik dark red (*Epipactis atrorubens*), northern daubing (*Galium boreale* L.), wild angelica (*Angelica sylvestris*), common golden rose (*Solidago virgaurea*), common bramble (*Rubus saxatilis* L.), arctic raspberry (*Rubus arcticus* L.). Moss cover. GPC 50–60%. Green mosses *Hylocomium splendens, Pleurozium schreberi, Dicranum elongatum* Schleich. ex Schwägr. | |
| | Lower part of the karst crater slope | |
| Site 5 | Spruce green-moss forest. Stand formula Picea 7 Betula 3. 63°20′18.7″ N, 52°54′11.5″ E, 145 m asl; slope 35–40°. | Gleyic gray-humus stratozem (Pantocolluvic Skeletic Stagnic Regosol (Abruptic, Ochric)). Soil profile has the AYao–RYg–D combination. |
| | Tree layer. Siberian spruce (*Picea obovata*) 13–15 m, birch downy (*Betula pubescens*) 5–7 m. Undergrowth. Birch (*Betula pubescens)* 3–5 m-high and Siberian spruce (*Picea obovata*) 2–4 m-high. Undergrowth. Willow filicifolia (*Salix phylicifolia* L.), Pallas honeysuckle (*Lonicera pallasii* Ledeb.) 1 m. Grass layer. GPC 10–15%. Herbs. Twisted pike (*Avenella flexuosa*) with PC 5–7%, common sagebrush (*Oxalis acetosella* L.), European sagebrush (*Lysimachia europaea*), * reed horsetail (*Equisetum scirpoides* Michx.), narrow-leaved willow-herb (*Epilobium angustifolium*), alpine sussurea (*Saussurea alpina* (L) DC.), marsh starwort (*Stellaria alsine* Grimm). Moss cover. GPC 100%. Mainly continuous cover of green mosses: *Pleurozium schreberi* (PC 30–40%) and *Ptilium crista-castrensis* (Hedw.) De Not. (PC 50–60%), also separate clumps of *Polytrichum commune* Hedw. with PC 10–15%. | |
| | Ukhta river valley, rock outcrops, slope | |
| | Upper part of the slope | |
| Site 6 | Mixed shrub–grass–broadleaf forest. Postfire. Forest stand formula Pinus 3 Picea 2 Betula 3 Populus 2 Larix[+]. 63°25′55.9″ N, 52°58′25.3″ E, 181 m asl | Gray-humus raw-humus podzolized residual-calcareous soil (Dystric Calcaric Skeletic Cambisol (Ochric, Nechic)). Soil profile has the O–A–Ye–AC–Cca–Rsa combination. |
| | Tree layer. Common pine (*Pinus sylvestris*) 17–19 m, Siberian spruce (*Picea obovata)* 17–19 m, downy birch (*Betula pubescens*) 15–17 m, trembling poplar or aspen (*Populus tremula*) 15–17 m, Siberian larch (*Larix sibirica)* 17–19 m. Undergrowth. Birch (*Betula pubescens)* 2–4 m, Siberian spruce (*Picea obovata)* 3–5 m, Scots pine up to 1.5 m. Undergrowth. prickly wild rose (*Rosa acicularis*) up to 1 m, common juniper (*Juniperus communis*) up to 1.5 m. Grass–bush layer. GPC 60–70%. Shrubs: blueberry (*Vaccinium myrtillus)* PC 40–45%. Herbs. Linnaean holodendron (*Gymnocarpium dryopteris* (L.) Newman) PC 10–15%, broad-leaved mayonnaise (*Maianthemum bifolium*), narrow-leaved willow-herb (*Epilobium angustifolium*), common golden rose (*Solidago virgaurea*), hairy oleaster (*Luzula pilosa* (L.) Willd.), wild angelica (*Angelica sylvestris*), small cow-wheat (*Melampyrum sylvaticum* L.), meadowfoam (*Melampyrum pretense* L.), twisted pike (*Avenella flexuosa*) PC 10–15%, wood geranium (*Geranium sylvaticum*), sheep fescue (*Festuca ovina*). Moss cover. GPC 10–15%. Green mosses *Hylocomium splendens, Pleurozium schreberi*. | |

**Table 2.** *Cont.*

| Sampling Sites | Plant Community, Plant Species | Soil Type |
| --- | --- | --- |
| | Lower part of the slope | |
| Site 7 | Sparse mixed herbaceous forest. Forest stand formula Pinus 4 Picea 1 Betula 3 Populus 2. 63°25′53.9″ N, 52°58′14.5″ E, 135 m asl | |
| | Tree layer. Common pine (*Pinus sylvestris*) 17–19 m, Siberian spruce (*Picea obovata*) 17–19 m, downy birch (*Betula pubescens*) 15–17 m, trembling poplar or aspen (*Populus tremula*) 15–17 m. Undergrowth. Birch (*Betula pubescens*), Siberian spruce (*Picea obovata*), common pine ( *Pinus sylvestris*), trembling poplar or aspen (*Populus tremula*). Undergrowth. Spikenard (*Rosa acicularis*) up to 1 m, common juniper (*Juniperus communis*) up to 1.5 m, Siberian knia (*Clematis alpina subsp. sibirica* (L.) Kuntze), and * cinnabarinus (*Cotoneaster cinnabarinus* Juz.). Grass–bush layer. GPC 70–80%. Shrubs: lingonberry (*Vaccinium vitis-idaea*), bilberry (*Vaccinium myrtillus*). Herbs. Small basilicum (*Thalictrum minus*), meadow grass (*Lathyrus pratensis* L.), northern clematis (*Galium boreale*), alpine sossurea (*Saussurea alpina*), tall larkspur (*Delphinium elatum* L.), common bramble (*Rubus saxatilis*), fence pea (*Vicia sepium* L.), round-leaved pear tree (*Pyrola rotundifolia* L.), wormwood (*Melica nutans*), hedgehog (*Dactylis glomerata* L.), * fragrant orchid (*Gymnadenia conopsea*), hairy oleaster (*Luzula Pilosa*), wild angelica (*Angelica sylvestris*), histote bitterling (*Polygala amarella*), wood geranium (*Geranium sylvaticum*), greater stitchwort (*Rabelera holostea* (L.) M. T. Sharples & E. A. Tripp), meadow bluegrass (*Poa pratensis*), sheep fescue (*Festuca ovina*), common golden rose (*Solidago virgaurea*), woodruff (*Melampyrum sylvaticum*), blueberry (*Polemonium caeruleum* L.), * whole-leaved godwit (*Tephroseris integrifolia*), common sleep grass (*Pulsatilla patens* (L) Mill.), european meadowsweet (*Trollius europaeus* L.), meadow grass (*Lathyrus pratensis*), sand violet (*Viola rupestris* F. Schmidt), germander speedwell (*Veronica chamaedrys* L.), * dremlik dark red (*Epipactis atrorubens*), carnation (*Dianthus superbus* L.), round-leaved bell (*Campanula rotundifolia* L.), buttercup (*Ranunculus polyanthemos* L.), finger sedge (*Carex digitata*). Moss cover. GPC 5–10%. Green mosses: *Rhytidiadelphus triquetrus* (Hedw.) Warnst. *Dicranum elongatum*. | Mucky–dark-humus carbolithozem (Calcaric Mollic Folic Leptosol). Soil profile has the NA–(Csa)–Rca combination. |

Note. PC—Projective coating; GPC—General projective coating. Calciphilic plants are marked with (*).

Forest plantations belonged to four formations: pine, spruce, birch, and aspen forests. Pine forests were the most widespread, confined to pine terraces of river valleys, tops and slopes of watershed hills, and sides of sinkholes. The ecotopic range of pine forests was well-represented according to the gradient of moisture accumulation: lichen pine forests—lichen green moss pine forests—green moss pine forests—sphagnum pine forests. All pine forests were of secondary origin, formed after logging. In the majority of the surveyed pine forests, traces of ground fires of varying age and intensity were registered. A typical forest community type, pine–bilberry–green-moss forest on the watershed surface, was considered as the background area. Mixed derived forests dominated by *Pinus sylvestris* and *Betula pubescens* were recorded on karst forms of relief, on the sides of sinkholes and river valley slopes.

Wetlands were not widely distributed in the study area but are an integral component of its landscapes and are characterized by a sufficiently high diversity. Hydromorphic communities covered approximately 10% of the surveyed area. Relatively large bogs and their systems of raised and transitional types are distributed in the watersheds. In the central part of the investigated territory, there are small-sized karstic raftings and key bogs. One of the background areas considered was a transitional sedge bog, located in the valley of the runoff of the forming creek.

*4.3. Oribatid Mites*

A total of 51 oribatid mite species from 39 genera and 31 families were found (Table 3). The largest number of species of oribatid mites was noted in forest phytocenoses: mixed motley grass forest, located in the upper part of the karst crater slope (S3), pine–bilberry–

green-moss forest in the depression between glacial hills and uplands (S2), also in mixed herb–grass–strawberry forest in the middle part of the karst crater slope (S4) (29, 26, and 22 species were found, respectively). The highest Shannon index is noted for the S3 site.

In these three forest communities, the largest abundance of oribatid mites was noted. In these three communities, there was a similar composition of the most abundant species. These were the species *Tectocepheus velatus, Suctobelbella acutidens,* and *Oppiella nova* in the S2 and S3, and also the species *Quadroppia quadricarinata.* The similarity of these three communities, in addition to the above *Q. quadricarinata* and *T. velatus*, also determined the epigeic species *Phthiracarus laevigatus, Hermannia scabra, Damaeus bituberculatus, Adoristes ovatus, Carabodes areolatus, Parachipteria punctata, Ceratozetes gracilis, Pergalumna nervosa*; hemiedaphic species *Euphthiracarus cribrarius, Heminothrus longisetosus*; and eurybiontic species *Oribatula tibialis.* These species are common in the taiga zone of the European North [32,65,66].

In the S5 site, located in the lower part of the karst crater, as well as in the forest communities of sites S3 and S4, located in the upper and middle part of the karst crater profile, representatives of the Oppiidae and Suctobelbidae families were dominant in abundance, such as *Rhinoppia subpectinata* and *S. acutidens*, as well as the species of family Quadroppiidae *Q. quadricarinata*. The species of Oppiidae, Suctobelbidae, and Quadropiidae families are representatives of the life form of the inhabitants of small soil wells, according to Krivolutsky [32]. Further, the most numerous species here included the eurybiontic species *S. laevigatus* and *T. velatus*.

Epigeic species—inhabitants of the soil surface and upper horizons of the litter, according to Krivolutsky [32]—predominated in the number of species in all forest sites: S3, S4, and S5. The most numerous representatives of this life form were the species *P. laevigatus, D. bituberculatus, C. gracilis, Neoribates aurantiacus*, and *P. nervosa*.

In site 6 (mixed shrub–grass–broadleaf forest, postfire, located on the upper part of the slope of Ukhta river valley, with rock outcrops), the inhabitants of small soil holes—*O. nova, Oppiella neerlandica, Q. quadricarinata, S. acutidens*; inhabitants of the soil surface and upper horizons of the litter—*D. bituberculatus* and *C. gracilis*; inhabitant of the litter layer—*H. peltifer*; and also eurybiontic species *T. velatus* and *Scheloribates laevigatus* were the most numerous.

In site 7 (in sparse mixed herbaceous forest, located in Ukhta river valley, on the lower part of the slope, with rock outcrops) the inhabitants of small soil wells—*O. nova, O. neerlandica,* and *S. acutidens*—were the most numerous. The total abundance of oribatids in this site was significantly lower than in site 6 and also than the forest communities on the slope of the karst sinkhole and in the pine–bilberry–green-moss forest, located in the depression between the glacial hills and uplands.

The most specific species composition of oribatid mites was in site 1—in sedge wetland community, located in the depression between the glacial hills and uplands. Hydrobiontic species *H. thienemanni, M. monodactylus*, and *M. foveolatus*, which were not observed in other sites, were found here. The first two species were dominant in abundance here, along with the eurybiontic species *Tectocepheus velatus*. Species *M. foveolatus* and *M. monodactylus* (as *Malaconothrus gracilis* Hammen, 1952) were noted earlier in the Ukhta district on raised bogs [67]. Moreover, the hemiedaphic species *Heminothrus peltifer*, which prefers habitats with high humidity, was found in site 1. The smallest number of species (nine) and the lowest abundance of oribatid mites are noted here (Table 3).

In general, the abundance of oribatid mites in site 2 (pine–bilberry–green-moss forest, located in the depression between the glacial hills and uplands) was statistically significantly higher compared with the other sites. The largest number of oribatid species was noted in site 3 (in mixed motley grass forest, located on the upper part of the karst crater slope).

According to the results of the ordination of the oribatid mite communities by the method of the NMDS, site 1—the sedge wetland community, located in the depression between the glacial hills and uplands—is separate from other sites (Figure 4). Along the X axis, one can observe the association of oribatid communities into two groups.

**Table 3.** The taxonomic composition and abundance (individuals/1 m$^{-2}$) of the oribatid mites in the studied plant communities.

| Taxon/Site | S1 | S2 | S3 | S4 | S5 | S6 | S7 | Life Form | Distribution |
|---|---|---|---|---|---|---|---|---|---|
| Brachychthoniidae (Thor, 1934) *Eobrachychthonius latior* (Berlese, 1910) | - | - | 500 ± 300 | - | - | - | - | nonspecialized | Holarctic |
| *Liochthonius (L.) sellnicki* (Thor, 1930) | - | - | - | - | - | 1700 ± 838.65 | - | nonspecialized | Holarctic |
| Hypochthoniidae (Berlese, 1910) *Hypochthonius rufulus* (Koch, 1835) | - | - | 900 ± 900 | - | - | - | - | nonspecialized | Semi cosmopolitan |
| Eulohmanniidae (Grandjean, 1931) *Eulohmannia ribagai* (Berlese, 1910) | - | - | 200 ± 200 | - | - | - | 500 ± 300 | nonspecialized | Holarctic |
| Euphthiracaridae (Jacot, 1930) *Euphthiracarus (E.) cribrarius* (Berlese, 1904) | - | 1100 ± 550.76 | 800 ± 489.90 | 1800 ± 1148.91 | - | - | 400 ± 163.30 | hemiedaphic | Holarctic |
| Phthiracaridae (Perty, 1841) *Phthiracarus (P.) laevigatus* (Koch, 1844) | - | 400 ± 400 | 4200 ± 2169.48 | 2600 ± 1290.99 | - | - | 1400 ± 416.33 | epigeic | Holarctic |
| *Phthiracarus (P.) globosus* (Koch, 1841) | - | - | 2100 ± 1279.32 | - | - | - | - | epigeic | Holarctic |
| Malaconothridae (Berlese, 1916) *Malaconothrus (M.) monodactylus* (Michael, 1888) | 4800 ± 1818.42 | - | - | - | - | - | - | hydrobiontic | Holarctic |
| *Malaconothrus (Trimalaconothrus) foveolatus foveolatus* (Willmann, 1931) | 2600 ± 2100.79 | - | - | - | - | - | - | hydrobiontic | Holarctic |
| Nothridae (Berlese, 1896) *Nothrus borussicus* (Sellnick, 1928) | - | - | - | 400 ± 400 | - | - | - | hemiedaphic | Holarctic |
| *Nothrus pratensis* (Sellnick, 1928) | - | 1400 ± 416.33 | - | 2400 ± 673.30 | - | - | 3000 ± 683.13 | hemiedaphic | Holarctic |
| Crotoniidae (Thorell, 1876) *Camisia (C.) biurus* (Koch, 1839) | - | - | 100 ± 100 | - | - | - | - | hemiedaphic | Holarctic |
| *Camisia (C.) biverrucata* (Koch, 1839) | - | - | - | 1000 ± 476.09 | - | - | - | hemiedaphic | Holarctic |
| *Camisia (C.) invenusta* (Michael, 1888) | - | 100 ± 100 | - | - | - | 900 ± 341.57 | - | hemiedaphic | Palearctic |
| *Heminothrus (H.) longisetosus* (Willmann, 1925) | - | 2900 ± 680.69 | 1100 ± 191.49 | 3300 ± 1112.06 | 1600 ± 673.30 | 1300 ± 300 | 3300 ± 1508.86 | hemiedaphic | Holarctic |
| *Heminothrus (Platynothrus) peltifer* (Koch, 1839) | 100 ± 100 | 400 ± 282.84 | - | 300 ± 191.48 | 600 ± 258.20 | 6100 ± 2374.17 | 600 ± 258.20 | hemiedaphic | Semi-cosmopolitan |
| Nanhermanniidae (Sellnick, 1928) *Nanhermannia (N.) sellnicki* (Forsslund, 1958) | - | 3300 ± 1619.67 | 500 ± 300 | - | - | 1300 ± 100 | - | epigeic | Palearctic |

**Table 3.** *Cont.*

| Taxon/Site | S1 | S2 | S3 | S4 | S5 | S6 | S7 | Life Form | Distribution |
|---|---|---|---|---|---|---|---|---|---|
| Hermanniidae (Sellnick, 1928) *Hermannia (Heterohermannia) scabra* (L. Koch, 1879) | - | 200 ± 200 | 1200 ± 516.40 | 100 ± 100 | - | - | - | epigeic | Holarctic |
| Damaeidae (Berlese, 1896) *Caenobelba compta compta* (Kulczynski, 1902) | - | 200 ± 200 | 2400 ± 2141.65 | - | - | 1200 ± 230.94 | 1400 ± 258.20 | epigeic | Palearctic |
| *Damaeus (Epidamaeus) bituberculatus* (Kulczynski, 1902) | - | 600 ± 258.20 | 2400 ± 711.81 | 900 ± 574.46 | 3100 ± 660.81 | 8200 ± 3039.74 | 1000 ± 382.97 | epigeic | Palearctic |
| Ceratoppiidae (Grandjean, 1954) *Ceratoppia bipilis bipilis* (Hermann, 1804) | - | - | - | - | - | 400 ± 282.84 | - | epigeic | Holarctic |
| Liacaridae Sellnick, 1928 *Adoristes (A.) ovatus* (Koch, 1839) | - | 500 ± 100 | 600 ± 258.20 | 200 ± 200 | 900 ± 525.99 | 1100 ± 191.49 | - | epigeic | Holarctic |
| *Liacarus (L.) xylariae* (Schrank, 1803) | - | - | 100 ± 100 | - | - | - | 200 ± 115.47 | epigeic | Palearctic |
| Eremaeidae (Oudemans, 1900) *Eueremaeus oblongus silvestris* (Forsslund, 1956) | - | - | 1700 ± 660.81 | - | - | 100 ± 100 | 1900 ± 1037.63 | epigeic | Palearctic |
| Thyrisomidae (Grandjean, 1954) *Banksinoma lanceolata* (Michael, 1885) | - | - | - | - | - | - | 600 ± 476.09 | euedaphic | Holarctic |
| Oppiidae (Sellnick, 1937) *Rhinoppia (R.) subpectinata* (Oudemans, 1900) | - | - | 32300 ± 5940.54 | 41800 ± 11686.46 | 39400 ± 7634.13 | - | - | euedaphic | Holarctic |
| *Microppia minus minus* (Paoli, 1908) | 2200 ± 886.94 | - | - | - | - | - | - | euedaphic | Cosmopolitan |
| *Oppiella (O.) nova* (Oudemans, 1902) | 3900 ± 1330.41 | 26900 ± 18119.88 | 10300 ± 2440.63 | - | - | 24300 ± 3582.83 | 17600 ± 3676.95 | euedaphic | Cosmopolitan |
| *Oppiella (Moritzoppiella) neerlandica* (Oudemans, 1900) | - | 27000 ± 8185.35 | - | - | - | 8100 ± 957.43 | 6500 ± 4196.43 | euedaphic | Holarctic |
| Quadroppiidae (Balogh, 1983) *Quadroppia (Q.) quadricarinata* (Michael, 1885) | - | 6700 ± 2573.58 | 6800 ± 1451.44 | 400 ± 230.94 | 5800 ± 1518.77 | 9900 ± 2568.40 | - | euedaphic | Semi-cosmopolitan |
| Suctobelbidae (Jacot, 1938) *Suctobelbella (S.) acutidens acutidens* (Forsslund, 1941) | - | 7400 ± 5144.58 | 10400 ± 2292.02 | - | 13600 ± 3153.83 | 6700 ± 1408.31 | 7200 ± 3794.73 | euedaphic | Holarctic |
| *Suctobelbella (S.) palustris* (Forsslund, 1950) | 900 ± 525.99 | - | - | 9800 ± 3126.23 | - | - | - | euedaphic | Holarctic |

**Table 3.** *Cont.*

| Taxon/Site | S1 | S2 | S3 | S4 | S5 | S6 | S7 | Life Form | Distribution |
|---|---|---|---|---|---|---|---|---|---|
| Carabodidae Koch, 1843 | | | | | | | | | |
| *Carabodes (C.) areolatus* (Berlese, 1916) | - | 600 ± 200 | 300 ± 191.49 | 100 ± 100 | - | - | - | epigeic | Holarctic |
| *Carabodes (C.) marginatus* (Michael, 1884) | - | - | - | - | - | 400 ± 163.30 | - | epigeic | Palearctic |
| Hydrozetidae (Grandjean, 1954) | | | | | | | | | |
| *Hydrozetes thienemanni* (Strenzke, 1943) | 4500 ± 1398.81 | - | - | - | - | - | - | hydrobiontic | Holarctic |
| Tectocepheidae (Grandjean, 1954) | | | | | | | | | |
| *Tectocepheus velatus* (Michael, 1880) | 7500 ± 2270.83 | 30300 ± 11959.24 | 3000 ± 621.83 | 7400 ± 2253.89 | 3800 ± 840.64 | 19000 ± 4795.83 | 2300 ± 1123.98 | eurybiontic | Cosmopolitan |
| Phenopelopidae (Petrunkevitch, 1955) | | | | | | | | | |
| *Eupelops plicatus* (Koch, 1835) | - | 100 ± 100 | - | 3400 ± 1290.99 | - | 1600 ± 588.78 | 1700 ± 754.98 | epigeic | Holarctic |
| Achipteriidae (Thor, 1929) | | | | | | | | | |
| *Parachipteria punctata* (Nicolet, 1855) | - | 1000 ± 115.47 | 2700 ± 550.76 | 1100 ± 550.76 | 1900 ± 640.31 | - | 500 ± 378.59 | epigeic | Holarctic |
| Ceratozetidae (Jacot, 1925) | | | | | | | | | |
| *Ceratozetes (C.) gracilis* (Michael, 1884) | - | 1500 ± 660.81 | 4700 ± 525.99 | 6100 ± 2253.15 | - | 19200 ± 5027.92 | 3300 ± 1247.66 | epigeic | Cosmopolitan |
| *Melanozetes mollicomus* (Koch, 1839) | - | - | - | - | 700 ± 472.58 | - | - | epigeic | Holarctic |
| *Fuscozetes fuscipes* (Koch, 1844) | - | - | 100 ± 100 | - | - | - | 600 ± 382.97 | epigeic | Holarctic |
| Chamobatidae (Thor, 1937) | | | | | | | | | |
| *Chamobates (C.) pusillus* (Berlese, 1895) | - | 1800 ± 757.19 | 2000 ± 832.67 | - | - | - | - | epigeic | Holarctic |
| Humerobatidae (Grandjean, 1971) | | | | | | | | | |
| *Diapterobates humeralis* (Hermann, 1804) | - | 400 ± 282.84 | - | - | - | - | - | epigeic | Holarctic |
| *Diapterobates oblongus* (L. Koch, 1879) | - | - | - | - | 700 ± 100 | 1100 ± 191.49 | 400 ± 282.84 | epigeic | Palearctic |
| Oribatulidae (Thor, 1929) | | | | | | | | | |
| *Oribatula (O.) tibialis* (Nicolet, 1855) | - | 900 ± 251.66 | 1900 ± 500 | 2600 ± 1465.15 | - | - | - | eurybiontic | Holarctic |
| *Oribatula (Zygoribatula) exilis* (Nicolet, 1855) | - | 200 ± 200 | 100 ± 100 | 100 ± 100 | - | - | - | eurybiontic | Holarctic |

**Table 3.** *Cont.*

| Taxon/Site | S1 | S2 | S3 | S4 | S5 | S6 | S7 | Life Form | Distribution |
|---|---|---|---|---|---|---|---|---|---|
| Liebstadiidae (Balogh et P. Balogh, 1984) | | | | | | | | | |
| *Liebstadia (L.) pannonica pannonica* (Willmann, 1951) | - | - | - | - | 1000 ± 757.19 | - | - | eurybiontic | Holarctic |
| Scheloribatidae (Grandjean, 1933) | | | | | | | | | |
| *Scheloribates (S.) laevigatus* (Koch, 1835) | - | 1300 ± 443.47 | - | 4600 ± 959.17 | 3700 ± 869.87 | 9600 ± 3398.04 | 3100 ± 660.81 | eurybiontic | Semi-cosmopolitan |
| Parakalummidae (Grandjean, 1936) | | | | | | | | | |
| *Neoribates (N.) aurantiacus* (Oudemans, 1914) | - | - | 2800 ± 516.40 | - | 3400 ± 774.59 | - | - | epigeic | Holarctic |
| Galumnidae (Jacot, 1925) | | | | | | | | | |
| *Galumna (G.) lanceata* (Oudemans, 1900) | 200 ± 115.47 | - | - | - | - | - | - | epigeic | Palearctic |
| *Pergalumna (P.) nervosa nervosa* (Berlese, 1914) | - | 600 ± 346.41 | 1800 ± 200 | 2500 ± 1075.48 | 2000 ± 230.94 | - | 1400 ± 683.13 | epigeic | Holarctic |
| Total species | 9 | 26 | 29 | 22 | 15 | 20 | 22 | | |
| total | [ag] 26,700 ± 5748.91 | [b] 117,800 ± 44,144.76 | [cb] 97,300 ± 9364.29 | [aemq] 92,900 ± 20,739.90 | [afc] 82,200 ± 9999.33 | [gknq] 122,200 ± 16,235.35 | [dbc] 58,300 ± 11,527.79 | | |
| H′ | 1.851 | 2.102 | 2.508 | 2.08 | 1.849 | 2.397 | 2.474 | | |
| Berger–Parker | 0.2809 | 0.2572 | 0.3296 | 0.4499 | 0.4793 | 0.1989 | 0.2988 | | |
| Life forms (number of species) | | | | | | | | | |
| epigeic | 1 | 13 | 16 | 9 | 7 | 10 | 11 | | |
| hemiedaphic | 1 | 5 | 3 | 6 | 2 | 3 | 4 | | |
| euedaphic | 3 | 4 | 4 | 3 | 3 | 4 | 4 | | |
| eurybiontic | 1 | 4 | 3 | 4 | 3 | 2 | 2 | | |
| nonspecialized | - | - | 3 | - | - | 1 | 1 | | |
| hydrobiontic | 3 | - | - | - | - | - | - | | |

Notes. S1—Site 1. Sedge wetland community. Depression between glacial hills and uplands. S2—Site 2. Pine–bilberry–green-moss forest. Depression between glacial hills and uplands. S3—Site 3. Mixed motley grass forest. Upper part of the karst crater slope. S4—Site 4. Mixed herb–grass–strawberry forest. Middle part of the karst crater slope. S5—Site 5. Spruce green-moss forest. Lower part of the karst crater slope. S6—Site 6. Mixed shrub–grass–broadleaf forest. Post-fire. Ukhta river valley, rock outcrops. Upper part of the slope. S7—Site 7. Sparse mixed herbaceous forest. Ukhta river valley, rock outcrops. Lower part of the slope. - The absence of a species. (a, b, c, d, e, f, m, n, g, k and q) significance of differences in the population density; Mann–Whitney U-test: $p < 0.05$.

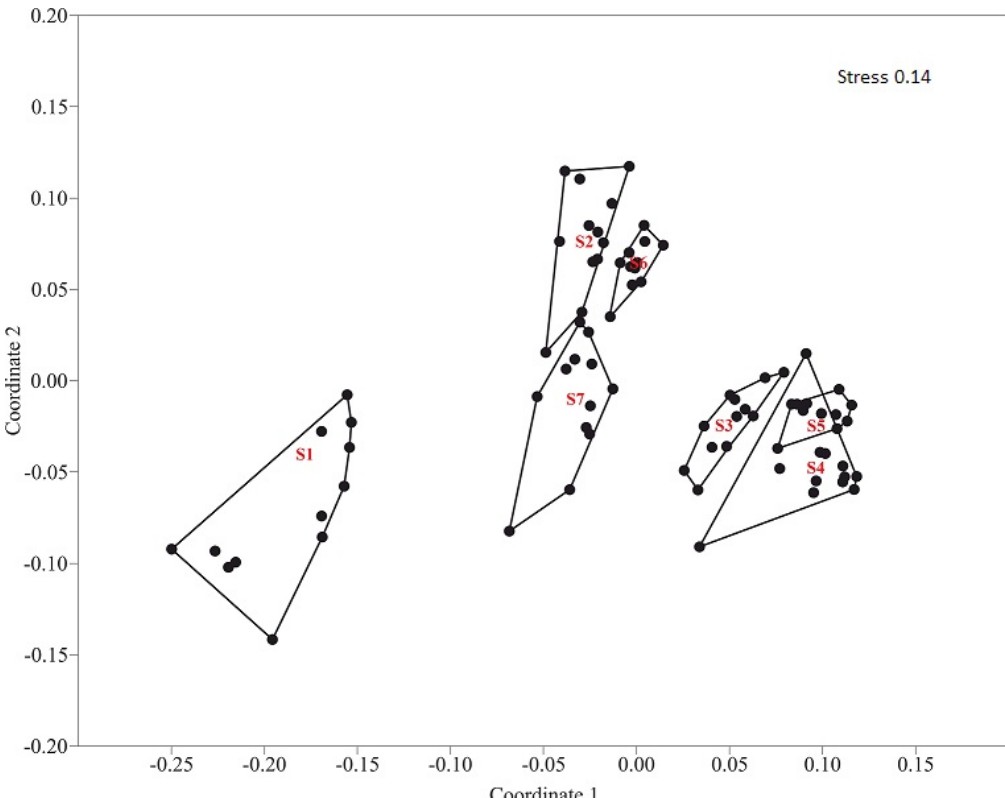

**Figure 4.** Ordination of communities of soil oribatid mites in the studied sites by the method of nonmetric multidimensional scaling (NMDS) using the Bray–Curtis index based on the relative abundance of their individual taxa.

- S1—Sedge wetland community. Depression between glacial hills and uplands.
- S2—Pine–bilberry–green-moss forest. Depression between glacial hills and uplands.
- S3—Mixed motley grass forest. Upper part of the karst crater slope.
- S4—Mixed herb–grass–strawberry forest. Middle part of the karst crater slope.
- S5—Spruce green-moss forest. Lower part of the karst crater slope.
- S6—Mixed shrub–grass–broadleaf forest. Postfire. Ukhta river valley, rock outcrops. Upper part of the slope.
- S7—Sparse mixed herbaceous forest. Ukhta river valley, rock outcrops. Lower part of the slope.

One of these groups includes sites S2, S6, and S7, which are the pine–bilberry–green-moss forest in the depression between glacial hills and uplands, mixed shrub–grass–broadleaf forest on the upper part of the slope, and sparse mixed herbaceous forest in the lower part of the slope with rock outcrops located Ukhta river valley, respectively. The second group comprises sites S3, S4, and S5, located on the slope of the karst crater, which has mixed motley grass forest on the upper part, mixed herb–grass–strawberry forest on the middle part, and spruce green-moss forest on the lower part of the karst crater slope, respectively. At the same time, each community inside group 1 and group 2 was allocated into a separate group. Species of oribatid mites, which created the specifics of each community, were noted.

The faunal composition was dominated by widespread Holarctic species (34 species, 66.7%). Cosmopolitan and semicosmopolitan made up 15.7% (eight species) in general. There were few Palearctic species (nine species, 17.6%). Most of the species were noted earlier in the taiga and tundra zones of the European North [65–70].

Some species were found that are rare in the taiga zone. They are common at lower latitudes. Species *Diapterobates oblongus* and *Diapterobates humeralis* in the European part of Russia are common in the mixed and broad-leaved forest zone, as well as in the zones of forest-steppe and steppe [32]. In the taiga zone, they are rare. In the taiga zone, we

found these species in epiphytic lichens, with a slight abundance [71]. At high latitudes (in the Arctic and Subarctic), another species of genus *Diapterobates* is common; these are *Diapterobates notatus* (Thorell, 1871), which are widespread and numerous in the Arctic Islands and Archipelagoes [72–74] and also found in the mainland tundra [68]. The species was noted in the Polar Urals [75].

*Phthiracarus laevigatus* was previously noted (as *Phthiracarus nitens* (Nicolet, 1855)) in the taiga zone (middle taiga subzone) on the territory of the Komi Republic [76]. On the territory of the European part of Russia, this species is distributed in zones of mixed and broad-leaved forests [32]. It was also noted in the zone of subtropical forests (in the vicinity of Sochi) in the Caucasus (Teberda) [77]. L. Subias indicates that this is a golarctic species, which, in the north of the Palearctic is not often found [60].

It can be concluded that along with widespread, polysonal species of oribatid mites in karst landscapes, several species were found that can be called "conditionally southern" species.

## 5. Conclusions

Soils, species composition of phytocenosis, and taxonomic diversity of oribatid mites in karst landscapes of the Timan Ridge, in the European northeast of Russia, in the conditions of northern taiga forests, are described. The research sites were located on a flat top of the hill, in the profile of the karst crater (in the lower, middle, and upper part of the crater), and on the slope profile in the Ukhta River Valley (on the upper and lower part of the slope). In total, seven sites have been considered.

On top of the hill, on fluvioglacial deposits, which are underlain on moraine loams, soil cover is subordinate to the zonal features; iron-illuvial contact-gley podzols (Stagnic Albic Rustic Podzols (Arenic)) are spread here. Part of the area is occupied by hydromorphic soils—Peaty eutrophic gley silty-peat soils (Eutric Rheic Hemic Histosols). Formation of such soils is closely connected with the functioning of wetland ecosystems. In the valleys of the river, soils are formed with the occurrence of Lower Perm deposits, represented by terrigenous-carbonate and sulfate formations; in karst funnels, they are formed with the underlying carbonate-terrigenous, red-colored thickness of the deposits of the Ufa tier. In the upper part of the slopes of the deep karst sinkholes, iron-illuvial podzols (Albic Rustic Podzols (Arenic)) are present; in the middle part, musky–peaty peat-lithozems (Dystric Rendzic Folic Leptosols); and in the lower part and at the bottom, gleyic gray-humus stratozems (Pantocolluvic Skeletic Stagnic Regosols (Abruptic, Ochric)). In the valley of the Ukhta River, on the tops and upper gentle parts of the hill slopes, with a deeper occurrence of carbonate rocks, are located gray-humus raw-humus podzolized residual-calcareous soils (Dystric Calcaric Skeletic Cambisols (Ochric, Nechic)). On the steep slopes of the hills, crushed stone and incompletely developed soils are formed with the strong influence of denudation and accumulative processes—Mucky–dark-humus carbolithozems (Calcaric Mollic Folic Leptosols). Soils with a close arrangement of underlying dense indigenous rocks in the profile and on steep slopes noticeably differ from zonal soils at the tops of the hills by the conditions of soil formation and their physical and chemical properties.

The structure and species composition of karst landscape phytocenoses also differs from the characteristics of the communities of typical glacial landscapes. Calciphilic plant species include *Cotoneaster cinnabarinus*, *Cypripedium calceolus*, *Gymnadenia conopsea*, *Epipactis atrorubens*, *Tephroseris integrifolia*, *Polygala amarella* and *Equisetum scirpoides*.

A total of 51 oribatid mite species from 39 genera and 31 families were found. The highest taxonomic diversity of oribatids was noted in forest phytocenoses located on the upper and lower part of the karst crater slope, and also on rock outcrops on the lower part of the slope in Ukhta river valley. Ordination of oribatid mite community by NMDS method showed the association of sites S3, S4, and S5 located on the slope of the karst crater in one group, and sites S6 and S7 located on a slope in the Ukhta River Valley and S2 (pine–bilberry–green-moss forest located in depression between glacial hills and uplands) in another group. The swamp community was located separately from other communities. In the forest communities located in the lower, middle, and upper part of

the karst crater profile (sites 3, 4, and 5), a similar composition of dominant species was observed. The specifics of the population of oribatid mites of karst landscapes was that along with the features of fauna, characteristic for zonal north-taiga forests (the predominance of polyzonal widespread species) were found "the conditionally southern" species, the main area of distribution of which is located in lower latitudes. These are the species *Phthiracarus (P.) laevigatus*, *Diapterobates humeralis*, and *Diapterobates oblongus*. The study provides the basis for future studies of poorly known oribatid mites of karst landscapes of Northern Europe.

**Supplementary Materials:** The following supporting information can be downloaded at: https://www.mdpi.com/article/10.3390/d14090718/s1, Figure S1: Soil profiles of the main observation points in karst landscapes of Middle Timan; Figure S2: Vegetation communities surveyed in karst landscapes of Middle Timan.

**Author Contributions:** Conceptualization, E.N.M.; methodology, E.N.M., V.A.K. and S.V.D.; investigation, E.N.M., V.A.K. and S.V.D.; writing—original draft preparation, E.N.M., V.A.K. and S.V.D.; writing—review and editing, E.N.M., V.A.K. and S.V.D.; Photos, soils, S.V.D., plant communities, V.A.K., oribatid mites, E.N.M. All authors have read and agreed to the published version of the manuscript.

**Funding:** This study was performed within the framework of budgetary themes: "The diversity of fauna and the spatial-ecological structure of the population animal of the European North-East of Russia and the neighboring territories in the context of environmental changes and economic development" (State registration number: 122040600025-2), "Assessment of ecological-coenotic, species and population diversity of the flora of key specially protected natural areas of the Komi Republic" (State registration number: 122040600026-9), "Cryogenesis as a factor of soil formation and evolution in arctic and boreal ecosystems of the European Northeast under current anthropogenic impacts, global and regional climatic trends" (State registration number: 122040600023-8).

**Institutional Review Board Statement:** Not applicable.

**Data Availability Statement:** Not applicable.

**Acknowledgments:** Sponsors took part in the design of the study and in the organization of field observations. Chemical analyzes of soil samples were carried out at the Ecosanalytic Laboratory of the Institute of Biology of the FIC Komi NC URA RAS "Ecoanalite". The Institute of Biology was provided with an instrumental base for processing the resulting material, including microscopes. The authors thank Antonina D. Adamova for the creation of a graphical abstract.

**Conflicts of Interest:** The authors declare no conflict of interest.

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
