# Peer review of "Karst Ecosystems of Middle Timan, Russia: Soils, Plant Communities, and Soil Oribatid Mites"

_diversity, doi:10.3390/d14090718_

Round 1
Reviewer 1 Report
The work is local, however, it was very well prepared by the authors.
There are no statistical parameters of the research. Have the physicochemical tests been repeated?
Reviewer 2 Report
In the study there is no any relationship, whereas it could be explored the relationships between mite diversity and plant community types. To do this;
1. Mite species diversity should be calculated using diversity indices such as Shannon index or Simpson diversity index.
2. Plant communities should be compared based on mite diversities on a graphic.
Some corrections and suggestions are marked on the text.
Best regards,

Author Response
Response to Reviewer 2
In the study there is no any relationship, whereas it could be explored the relationships between mite diversity and plant community types. To do this;
Point 1:
Mite species diversity should be calculated using diversity indices such as Shannon index or Simpson diversity index.
Response 1:
Diversity indices added to Table 3: Shannon Index and Berger-Parker Index for each site.
Point 2:
Plant communities should be compared based on mite diversities on a graphic.
Response 2:
Now the characteristics of the Oribatid communities in various plant communities are given. A comparison of plant communities based on the diversity of oribatid mites will be presented graphically.
For comments and additions made in the text of the manuscript, see the answers in the notes to the text of the manuscript.
See the attached file.

Reviewer 3 Report
See attachment.

Reviewer 4 Report
General comments
The manuscript, titled “Karst ecosystems of Middle Timan: soils, plant communities, and soil invertebrates”, describes the soil characteristics, phytocenose species composition and taxonomic diversity of oribatid mites in a mosaic karst landscape in the Timan Ridge, North-East European Russia.
The study is conducted in a unique and highly interesting ecosystem in terms of soil characteristics, plant communities and oribatid taxonomic diversity. As an oribatid mite researcher, I will mostly focus on the parts of the manuscript that deal with oribatid mites in my review.
The oribatid mite study is based on a quite large dataset, and the number of oribatid mite specimen (10200) identified is impressive. However, the paper is very descriptive in its current form and could be improved. I would encourage the authors to add statistical analyses (eg. biodiversity indices) and also more information and discussion regarding oribatid mite diversity, ecology and/or other relevant topics. It seems to me that the data collected could provide much more, that there are interesting results and outcomes, which are not presented in the manuscript in its current form. Please see more specific comments below.
I suggest the manuscript to be accepted for publication in Diversity after moderate revision.
Specific comments
Title
Please consider using “oribatid mites” instead of “soil invertebrates” as oribatid mites were the target group of this study.
Introduction
Characteristics and development of karst landscapes are well explained in the Introduction. On the other hand, the aims of the study are described in one sentence only, and I would encourage the authors to describe the aims of the study in a bit more detail. As a reader I would like to know, why this study is important. Maybe because karst landscapes in general have high species diversity and/or the oribatid mite communities in this specific karst ecosystems in Middle Timan have not been studied before? However, I am not sure if this is needed because the manuscript will be part of a special issue concentrating on karst landscapes.
There is very little information about oribatid mites in the Introduction. Reader is left wondering: Why oribatid mites were used as representative species? Maybe because of their high diversity or indicator potential? I would encourage the authors to add more information of Oribatida, regarding their diversity, indicator values, ecological meaning and/or the fact that they are found almost everywhere in the world and from almost all kinds of ecosystems etc.
Material and methods
Study area and the procedure of collection of research material are well described. Number of samples collected (12 samples from each 7 sites) to study oribatid mite communities is appropriate. The number of oribatid specimen (10200) identified from the samples is impressive. However, I would encourage the authors to conduct biodiversity index calculations or other statistical analysis on their data. Please consider doing that.
Results and discussion
The section that describes the soil is very detailed and comprehensive. At the same time, the section that deal with oribatid mite communities is very short. I think the manuscript could be improved by further analysis of the oribatid mite data, and by adding more discussion on the findings. For example, the authors note: “The highest taxonomic diversity of oribatid mites was noted in forest phytocenoses located in the upper part of the karst crater slope, the lower part of the karst crater slope, also on rock outcrops in the lower part of the slope in Ukhta river valley; 18, 17 and 17 species were found here, respectively” (please see lines 475-478).
Were these differences significant? What the authors think about the differences, why only 8 species were found from the site 1 (maybe because it is a wetland site?), and is there an explanation why 11 species were recorded from Pine-bilberry-green-moss forest and 18 from Mixed motley grass forest (please see Table 3). Why biodiversity indices were not calculated? How about oribatid abundance, did it differ between the seven study sites? How do oribatid mite species diversity relate to soil characteristics or landscape mosaicism? The authors have collected good dataset, which I think could provide much more information on oribatid mites than is presented in the manuscript in its current form.
On lines 479-482, the authors write: “Most of the species were noted earlier in the taiga and tundra zones of the European North [33-37]. At the same time, some species should be paid attention to, these are species Phthiracarus (P.) laevigatus (Koch, 1844), Banksinoma lanceolata (Michael, 1885), Fuscozetes fuscipes (Koch, 1844), Diapterobates oblongus (L. Koch, 482 1879), which are not often found in the taiga zone of the European North-East.”
This discussion could be extended. For example: In which kind of habitats these species live? How about the oribatid mite communities in karts ecosystems, are they characterized by generalist or specialist species? Which species had the highest abundance, and which were found in low numbers?
In general, the oribatid mite section of the manuscript is very short. I think the article could be improved remarkably if the authors added more analyses and discussion regarding the oribatid mites and their diversity and ecology. Please consider doing that.
Conclusions
There is nothing about the oribatid mite or vegetation communities in the conclusion section. It is stated in the introduction in lines 58-60 that: “The aim of this study was to determine the diversity of soil types, plant communities, and soil invertebrates in karst relief forms under conditions of North taiga forests using oribatid mites as a representative species.” This is not in line with the conclusions section, so I suggest the authors to add concluding remarks regarding oribatid and plant communities.
References
I would suggest to add, along with more information on oribatid mites, references to recent studies regarding oribatid mite diversity, ecology etc. Majority of the references to oribatid mite studies seem to be author’s own works. These citations are relevant as they deal with Russian oribatid mite diversity, however, I encourage the authors to add more references to other oribatid mite studies as well.
Reviewer 5 Report
The presented article is a simple inventory of the ecosystem. The article lacks the research problem and the statistical analysis of the results. It is not suitable for publication in the present form.
Round 2
Reviewer 2 Report
Dear authors,
Thanks to the authors for the corrections on their revised manuscript.
The corrections I mentioned on the earlier version of the manuscript have been made by the authors. However, in the revised version, the number of species of oribatid mites has been increased from 35 to 51. Since you didn't do the rework, this had to be needed to be explained.
Author Response
Dear Reviewer,
Thank you for your attention to our manuscript and for your comments.
The time we received to revise the manuscript made it possible to carry out identifications to the species level of some individuals that could not be previously identified. Additional work carried out made it possible to expand the list of oribatid mites in the studied sites. As a result, the list now includes 51 species of oribatid mites.
With respect,
Elena N. Melekhina
Reviewer 4 Report
I think the ms has improved a lot as the authors have supplemented more information and discussion on oribatid mites, calculated diversity indices and abundances of oribatida. However, I would still encourage the authors to do statistical analyses on the abundance/diversity data. The question whether abundance or diversity of oribatid mites differed significantly between the sites, is still left unanswered. The ms is still quite descriptive in its current form.
Author Response
Dear Reviewer,
Thank you very much for your attention to our manuscript and for your valuable comments.
We calculated the significance of differences in the population density of oribatid mites was determined using the Mann–Whitney U-test at a significance level of p < 0.05; PAST software was used for data processing (Hammer et al., 2001).
In addition, the ordination of communities of soil oribatid mites in the studied sites by the method of non-metric multidimensional scaling (NMDS) using the Bray-Curtis index based on the relative abundance of their individual taxa was carried out. Ordination was carried out using the PAST program, the results are shown in Figure 4.
With best regards,
Elena N. Melekhina
Reviewer 5 Report
The Authors made some efforts to improve the mesofaunal analyzes. However, in my opinion, the analyzes do not meet the requirements set out in scientific journals.
Methods:
Characteristic of the study areas – The map showing the study sited and experimental design is missing
Please add the formulas, which were used for the calculation of ecological indices
Results:
The statistical data analysis is missing. At least the community responses should be compared using statistical methods.
Table 3 is difficult to analyze. Please consider, if some of the data should not be moved to supplementary material.
Discussion:
There is almost no discussion of the data. Those few sentences included in the results section, this is not a true discussion.
Conclusions:
The conclusions are not clear. Please specify the main findings of this article
Author Response
Dear Reviewer,
Thank you very much for your attention to our manuscript and for your valuable comments.
See the answer to the remarks of the reviewer in the attached file.
With best regards,
Elena N. Melekhina

Round 3
Reviewer 2 Report
Dear author/editor,
The author's answer is appropriate.
The manuscript has been considerably improved according to reviewers’ comments and can be acceptable in its current form.
Best regards,
Author Response
Dear reviewer,
Thank you very much for your attention to our work and your positive decision. Your comments helped to significantly improve the quality of our paper.
Sincerely,
Elena N. Melekhina